# Parameter tuning and model selection in Optimal Transport with semi-dual Brenier formulation

**Adrien Vacher**
LIGM, Univ. Gustave Eiffel, CNRS
MOKAPLAN, INRIA
`adrien.vacher@univ-eiffel.fr`

**François-Xavier Vialard**
LIGM, Univ. Gustave Eiffel, CNRS
`francois-xavier.vialard@univ-eiffel.fr`

## Abstract

Over the past few years, numerous computational models have been developed to solve Optimal Transport (OT) in a stochastic setting, where distributions are represented by samples and where the goal is to find the closest map to the ground truth OT map, unknown in practical settings. So far, no quantitative criterion has yet been put forward to tune the parameters of these models and select maps that best approximate the ground truth. To perform this task, we propose to leverage the Brenier formulation of OT. Theoretically, we show that this formulation guarantees that, up to sharp a distortion parameter depending on the smoothness/strong convexity and a statistical deviation term, the selected map achieves the lowest quadratic error to the ground truth. This criterion, estimated via convex optimization, enables parameter tuning and model selection among entropic regularization of OT, input convex neural networks and smooth and strongly convex nearest-Brenier (SSNB) models. We also use this criterion to question the use of OT in Domain-Adaptation (DA). In a standard DA experiment, it enables us to identify the potential that is closest to the true OT map between the source and the target. Yet, we observe that this selected potential is far from being the one that performs best for the downstream transfer classification task.

## 1 Introduction

Optimal transport (OT) is a tool to compare probability distributions that has found numerous applications ranging from economics [Galichon, 2016, Chiappori et al., 2010], unsupervised learning [Sim et al., 2020], shape matching [Feydy et al., 2017], NLP [Chen et al., 2019, Alvarez-Melis and Jaakkola, 2018] and biology [Schiebinger et al., 2019, Tong et al., 2020]. In its dual form, OT is a linear maximization problem on functions, which are called potentials, subject to a cost constraint. When the cost is chosen to be quadratic, the solutions of this problem are convex and their gradient provide optimal maps that transport one distribution onto the other. In a significant part of the OT applications, the transport map itself is the object of interest. For instance in Domain-Adaptation, the source distribution is transported on the target [Courty et al., 2017], for color transfer one color histogram is transported on the other [Rabin et al., 2014] and in biology, the RNA cell expression profile is interpolated in time using OT maps [Schiebinger et al., 2019]. Over the past few years, many models and computational methods [Cuturi, 2013, Genevay et al., 2016, Seguy et al., 2018, Bonneel and Coeurjolly, 2019, Vacher et al., 2021] were proposed and implemented to estimate these optimal transport maps. Under regularity assumptions, some of these models were shown to accurately estimate the original transport map provided the models use optimal parameters [Pooladian and Niles-Weed, 2021, Manole et al., 2021]. When such results exist, either the parameters to use are explicit but they are impractical as they rely on generic worst-case bounds, either they involve unavailable constants. To the best of our knowledge, no quantitative criterion has yet been devised to tune the parameters of OT models and later discriminate between calibrated models. The setting

36th Conference on Neural Information Processing Systems (NeurIPS 2022).

we are interested in is standard in statistical and ML applications, in which probability measures are only accessible via samples in Euclidean spaces. The goal is to recover, for the quadratic cost, a potential chosen among different models/parameters that is the closest to the unknown ground truth. For achieving this task, we put forward the use of the semi-dual functional of OT that we now define.

**The semi-dual (Brenier) objective.** The quantitative criterion that we propose is the so-called *semi-dual* Brenier objective of OT. It is a convex functional on the space of functions $f : \mathbb{R}^d \to \mathbb{R}$ and is defined for $\mu, \nu$ two probability measures on the Euclidean space by, denoting $\langle \cdot, \cdot \rangle$ the pairing between Radon measures and continuous functions, $J_{\mu,\nu}(f) := \langle f, \mu \rangle + \langle f^*, \nu \rangle$ with $f^*$ the Fenchel-Legendre transform of $f$ given by $f^*(y) := \sup_{x \in \mathbb{R}^d} x^\top y - f(x)$. Note that for a general cost $c$, this new objective can be obtained by replacing one potential by its $c$-transform in the Kantorovitch dual formulation. Now, whenever $f$ is convex, its legendre transform can be computed pointwise with an arbitrary precision. Furthermore, if $\nu$ is a finite sum of Dirac masses, the resulting convex problems of $J(f)$ are independent and can be solved in parallel.

**Related works.** The problem of evaluating OT models was recently studied by Korotin et al. [2021]. They proposed to generate synthetic ground truth optimal maps using an input convex neural network. Then, they calibrate various OT models on these ground truth OT maps and compare the performance of each calibrated OT model by measuring the natural $L^2$ distance between the estimated map and the ground truth. Their paper gives an interesting perspective on comparing current OT models. However, their setup requires the knowledge of the ground truth to calibrate the OT models. This limitation can be overcome for convex potentials as shown in our work.

The use of the Fenchel-Legendre transform can be found in the pioneering paper Brenier [1991]. It can be shown that this new formulation retains more convexity than the Kantorovitch formulation. On the theoretical side, this gain was leveraged for uses as diverse as sharp bounds for the problem of statistical map estimation [Hütter and Rigollet, 2021] or quantitative stability results of the transport map with respect to the measures [Delalande and Merigot, 2021]. On the numerical side, since the Fenchel-Legendre transform has linear cost on a grid (Fast Legendre Transform), the semi-dual is used to design efficient numerical algorithms in low dimension [Jacobs and Léger, 2020]. In the machine learning community, the semi-dual was proposed by Taghvaei and Jalali [2019] to estimate convex transport maps parametrized by Input Convex Neural Networks [Amos et al., 2017]. When $n$ is the sample size, they noticed that instead of the classical $O(n^2)$ complexity of OT, this new formulation leads to an $O(n)$ complexity per iteration as it only requires $n$-independent computations of the Legendre transform.

**Our contributions.** The goal of our paper is to answer the following question: given $(f_1, \cdots, f_p)$, $p$ convex potentials, how to select the one that minimizes the quadratic error, denoting $\nabla f_i$ the gradient of $f_i$, $e_\mu(f_i) = \int_x \|\nabla f_i(x) - T_0(x)\|^2 d\mu(x)$, where $T_0$ is the true, unknown, OT map from $\mu$ to $\nu$? A concrete example of this problem would be: we train a Sinkhorn model [Cuturi, 2013] on samples $(\hat{\mu}_{train}, \hat{\nu}_{train})$ with different temperatures $(\varepsilon_1, \cdots, \varepsilon_p)$, giving us empirical potentials $(\hat{f}_{\varepsilon_1}, \cdots, \hat{f}_{\varepsilon_p})$. Given test samples $(\hat{\mu}_{test}, \hat{\nu}_{test})$ how to choose the Sinkhorn empirical potential that minimizes the *unknown* error $(e_\mu(\hat{f}_{\varepsilon_i}))_{1 \le i \le p}$? We give an illustration of this problem in Fig. 1.

The main contribution of the paper is to use the empirical semi-dual to answer this question; to the best of our knowledge, it is the first time that the model selection problem in OT has been addressed. From a theoretical point of view, in the strongly-convex and smooth setting, we prove that the potential $f_i$ minimizing the empirical semi-dual is the one minimizing $e_\mu$ up to a sharp multiplicative factor that depends on the smoothness of the potential and up to an additive statistical deviation term. From an experimental point of view, we showcase on three synthetic experiments, with an ICNN model [Taghvaei and Jalali, 2019], a Sinkhorn model [Cuturi, 2013] and a SSNB model [Paty et al., 2020], that the potential achieving the lowest quadratic error is nearly always selected by the semi-dual. This consistent behavior enables

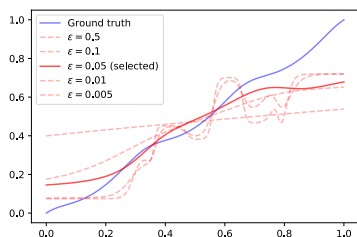

Figure 1: Parameter tuning for stochastic entropic OT: how to pic $\varepsilon_i$ that best fits the *unknown* blue curve?

us to conclude on a concrete ML application of our work: in Domain Adaptation, a widely accepted approach is to seek for an OT map between a source and a target using various models/parameters

in order to transfer the source's labels on the *unknown* target's labels. The question whether the OT model is truly relevant for this task has remained unanswered because of the impossibility to discriminate between candidates close to the "true" OT map and those "far" from it. In our last experiment, we bring a partial negative answer to this question: we observe that the map minimizing the semi-dual, hence the closest to the ground truth OT map, is often far from being the one that achieves the best label transfer.

**Assumptions and notations.** In this paper $X, Y$ are compact subsets of $\mathbb{R}^d$, $\mu$ and $\nu$ are probability measures over $X$ and $Y$ respectively with their $n$-samples empirical counterparts $\hat{\mu}$, $\hat{\nu}$. We shall denote by $\text{supp}(\mu)$, $\text{supp}(\nu)$ the support of $\mu$ and $\nu$ respectively. The cost we consider is the quadratic cost $c(x, y) = \|x - y\|^2/2$ and shall denote $q$ the quadratic function $q(x) = \|x\|^2/2$. Finally, we shall call $M$-smooth any function with an $M$-Lipschitz gradient.

## 2  Brenier formulation of OT

In its dual formulation, the OT problem optimizes over a pair of continuous functions, called *Kantorovitch* potentials $(\phi, \psi)$ subject to a cost constraint as $\text{OT}_c(\mu, \nu) = \sup_{(\phi, \psi)} \langle \phi, \mu \rangle + \langle \psi, \nu \rangle + \iota(\phi \oplus \psi \leq c)$ where $\phi \oplus \psi$ is defined as $(\phi \oplus \psi)(x, y) = \phi(x) + \psi(y)$ and $\iota$ is the convex indicator function. When $c$ is the Euclidean squared distance, we simply denote it by OT. In this case, if one of the two measures has density w.r.t. the Lebesgue measure, Brenier's theorem [Brenier, 1991] shows that a unique optimal map sending $\mu$ to $\nu$ exists and is given by the gradient of a convex function. If one further assumes regularity of the underlying densities and convexity of the support of the distributions, the optimal map gains in regularity.

**Theorem 1** (Caffarelli [2000] Theorem 2b.)**.** *Assume that $\mu$ and $\nu$ have $C^1$ densities bounded from below and above. If $\mu$, $\nu$ have compact and convex support, then, defining the Brenier potentials $(f, g) = (q - \phi, q - \psi)$, $f$ and $g$ are $C^2$ convex functions such that $\nabla f_\#(\mu) = \nu$ and $\nabla g_\#(\nu) = \mu$, where $T_\#(\eta)$ is the pushforward of the distribution $\eta$ by the map $T$ defined as $T_\#(\eta)(A) = \eta(T^{-1}(A))$ for all Borel $A$.*

Hence under the assumptions of Theorem 1, the optimal Brenier potentials are both smooth. In particular, since they are mutual Legendre transform of one and other, they are both also strongly convex.

**The semi-dual Brenier objective for convex potential selection.** In the context of statistical OT, the transport Kantorovitch potentials $\hat{\phi}, \hat{\psi}$ are usually estimated using the dual formulation of OT. To compare the obtained potentials, one may be tempted to simply evaluate the Kantorovitch linear objective on a test set $K_{\hat{\mu}, \hat{\nu}}(\hat{\phi}, \hat{\psi}) = \langle \hat{\phi}, \hat{\mu} \rangle + \langle \hat{\psi}, \hat{\nu} \rangle + \iota(\hat{\phi} \oplus \hat{\psi} \leq c)$ where $\hat{\mu}, \hat{\nu}$ represent independent samplings of $\mu, \nu$. However in numerous OT models, the learned potentials $(\hat{\phi}, \hat{\psi})$ usually do not respect the cost constraint and the $\iota(\hat{\phi} \oplus \hat{\psi} \leq c)$ term diverges. For instance, in the entropic regularization of OT the constraint is "loosely" satisfied on the train set since it replaces the hard inequality constraint by the soft penalization $\varepsilon \langle e^{\frac{\phi \oplus \psi - c}{\varepsilon}}, \hat{\mu} \otimes \hat{\nu} \rangle$. It is possible though to remove the cost constraint from the objective in order to evaluate candidate potentials. Rewriting the Kantorovich dual with the Brenier potentials gives $\text{OT}(\mu, \nu) = \langle q, \mu + \nu \rangle - \inf_{(f, g)} \langle f, \mu \rangle + \langle q, \nu \rangle + \iota(f(x) + g(y) \geq x^\top y)$ where the optimization is done on $f, g \in C^0$, the space of continuous functions. The inequality constraint implies that $g(y) \geq \sup_x x^\top y - f(x)$ for every $x$ which shows that we can replace $g$ by $f^*$, the Fenchel-Legendre transform of $f$. Therefore, up to moment terms, we get the *semi-dual* Brenier formulation $\inf_f J_{\mu, \nu}(f) = \langle f, \mu \rangle + \langle f^*, \nu \rangle$. This new nonlinear objective gains in convexity with respect to the Kantorovitch formulation (see Sec. 3) and now, whenever $f$ is strongly convex, $J_{\mu, \nu}(f)$ is finite and can be efficiently computed on discrete measures using standard convex optimization algorithms. Hence, if we restrict ourselves to convex $f$, possibly regularized with the addition of a small quadratic term, we are provided with a tractable and well-behaved selection criterion: the potential that minimizes $J_{\hat{\mu}, \hat{\nu}}$. We show in the next section that, thanks to the extra convexity, the minimization of this objective coincides, up to a stochastic term, with the minimization of the quadratic error $e_\mu(f) = \int_X \|\nabla f(x) - T_0(x)\|^2 d\mu$.

# 3  Potentials selection

We give in this section our main theoretical guarantee on model selection via the semi-dual. The setting we need to consider is the case of $M$-smooth and $\gamma$-strongly convex potentials, assumption which is discussed below. Let us first recall a stability result which shows that the semi-dual formulation is upper-bounded and lower-bounded by the quadratic error $e_\mu$. When no confusion is possible, we shall from now on denote $J_{\mu,\nu}$ by $J$.

**Lemma 1.** *Assuming that an optimal convex potential $f_0$ such that $T_0 = \nabla f_0$ pushes $\mu$ onto $\nu$ exists, then if $f$ is $M$-smooth, denoting $J_0 = J(f_0)$, we have $\frac{1}{2M} e_\mu(f) \leq (J(f) - J_0)$ where $e_\mu(f) = \|\nabla f - T_0\|^2_{L^2(\mu)}$. Conversely, if $f$ is a continuous $\gamma$-strongly convex function then $J(f) - J_0 \leq \frac{1}{2\gamma} e_\mu(f)$.*

This result is derived from Muzellec et al. [2021, Proposition 1], see Appendix for more details. A similar result can be found in Hütter and Rigollet [2021, Proposition 10] in a slightly less general form and in Makkuva et al. [2020, Theorem 3.6]. Importantly, note that there is no smoothness assumption made on the optimal map $T_0$. In contrast, we prove that the smoothness and strong-convexity assumptions on the candidate $f$ are necessary to lower and upper bound $J(f) - J_0$ with respect to $e_\mu$.

**Proposition 1.** *Take $\mu \sim \mathcal{U}([-\frac{1}{2}, \frac{1}{2}])$ and $f_0$ of the form $\lambda q(x) + x$ with $\lambda \geq 0$. The potential $f(x) = x$ is indeed convex with Lipschitz gradient and is such that $J(f) = +\infty$ yet $e_\mu(f) = \frac{\lambda^2}{4} \to 0$. Conversely, define the potential $g_0(x) = |x| + q(x)$ and for $0 \leq \lambda \leq \frac{1}{2}$, define $g_\lambda = g_0(\cdot - \lambda)$ which is indeed strongly convex (and even locally Lipschitz). The difference of semi-duals reads $J(g_\lambda) - J(g_0) = 2\lambda^2 - \lambda^3 \underset{\lambda \to 0}{\sim} 2\lambda^2$ and yet $e_\mu(g_\lambda) = 4\lambda + 5\lambda^2 \underset{\lambda \to 0}{\sim} 4\lambda$.*

The detailed computations are left in Appendix. Equipped with this lemma, we can derive our result.

**Proposition 2.** *Let $(f_1, \cdots, f_p)$ be $p$ potentials and $(\hat{\mu}, \hat{\nu})$ be the $n$-samples empirical counterparts of $(\mu, \nu)$. Let $i_0$ be the index of the map that minimizes the empirical semi-dual, $i_0 = \arg\min_i J_{\hat{\mu},\hat{\nu}}(f_i)$ and similarly $i_1 = \arg\min_i e_\mu(f_i)$. If $f_{i_0}$ is $M$-smooth and $f_{i_1}$ is $\gamma$-strongly convex and if an OT map from $\mu$ to $\nu$ exists, then for all $0 < \delta < 1$ we have with probability at least $1 - \delta$*

$$e_\mu(f_{i_0}) \leq \frac{M}{\gamma} e_\mu(f_{i_1}) + 8MC\sqrt{\frac{\ln(4/\delta)}{2n}}, \tag{1}$$

*where $C = \max(C_{i_0}, C_{i_1})$ with $C_i = \max(\|f_i\|_{X,o}, \|f_i^*\|_{Y,o})$ and $\|.\|_{Z,o}$ is defined as $\|g\|_{Z,o} = \sup_{z \in Z} g(y) - \inf_{z \in Z} g(y)$.*

The proof is left in Appendix and is a direct application of the previous stability estimate and the Hoeffding lemma. The following proposition shows that in the non-stochastic regime $n = +\infty$, our bound $\frac{M}{\gamma}$ is tight.

**Proposition 3.** *Take $\mu \sim \mathcal{U}[0,1]$, $f_0 \equiv 0$, $g(x) = Mq(x) + x$ and for $\epsilon > 0$, take $h_\epsilon(x) = \gamma q(x) + (\epsilon + \alpha_{M,\gamma})x$ with $\alpha_{M,\gamma} = \frac{\gamma}{2}\left[\sqrt{1 + \frac{4(M-\gamma)}{3\gamma}} - 1\right]$, and where we assumed $M \geq \gamma$. The potential $g$ is indeed $M$-smooth and $h$ is indeed $\gamma$-strongly convex and we have $\frac{e_\mu(g)}{e_\mu(h_\epsilon)} \underset{\epsilon \to 0}{\to} \frac{M}{\gamma}$ yet $J(g) - J(h_\epsilon) = -\frac{\epsilon}{2\gamma}(\epsilon + 2\alpha_{M,\gamma}) < 0$.*

The computations are left in Appendix. The bounds of Proposition 2 being tight in the non-stochastic setting, it is natural to question the assumptions on the class of potentials, namely $M$-smoothness and $\gamma$-strong convexity. Let us first mention Theorem 1 readily ensures that in the case where $\mu$, $\nu$ have $C^1$ densities with convex compact support then the optimal potentials satisfy the smoothness assumptions. Under such conditions on the measures, one should indeed benchmark potentials which are already smooth and strongly convex. However in practice, we may not have access to potentials satisfying these conditions. A simple idea to satisfy the smoothness assumption is to use the Moreau-Yosida transform $M_\tau(f) = \inf_y f(y) + \frac{1}{\tau} q(x - y)$, which verifies $f - \frac{L^2\tau}{2} \leq M_\tau(f) \leq f$ where $L$ is the Lipschitz constant of $f$ on $\nu$. The semi-dual shall read $\frac{1}{2\tau} e_\mu(M_\tau(f)) \leq J(M_\tau(f)) - J_0$, however without further assumptions, we cannot upper-bound

the original error $e_\mu(f)$ as $|e_\mu(M_\tau(f)) - e_\mu(f)|$ may not be in $o(1)$. The case of strong-convexity has a more favorable behavior. Denoting $Q_\delta(f) = f + \delta q$, we can apply Lemma 1 and obtain $J(Q_\delta(f)) - J_0 \leq \frac{1}{2\delta} e_\mu(Q_\delta(f))$. The error can be upper bounded by $e_\mu(f) + 4\delta^2 \langle q, \mu \rangle$. Under a local Lipschitz assumption on $f^*$, $J(Q_\delta(f)) - J_0$ is $O(\delta)$, hence we observe there is a trade-off between the deterioration of the factor in $O(\frac{1}{\delta})$ and the bias in $O(\delta)$. As a consequence, we can derive a new stability result, proved in Appendix, when the optimal $\delta$ is chosen. The strong-convexity assumption is relaxed with a local Lipschitz assumption of the conjugate at the cost of loosing on exponent on the error.

**Proposition 4.** *If $f$ is s.t. $f^*$ is $L$-Lipschitz on the support of $\nu$ then the semi-dual reads $J(f) - J_0 \leq 2\sqrt{e_\mu(f)[\frac{L^2}{2} + \langle q, \mu \rangle]}$.*

Note that at optimality, $f_0^*$ is indeed Lipschitz on the support of $\nu$, yet this condition may be difficult to enforce in practice. We believe though that sharper/less restrictive stability results could be obtained via a refined analysis and we postpone this interesting question for future works.

## 4  Sinkhorn potentials

The Sinkhorn model [Cuturi, 2013], defined as $S_\varepsilon(\mu, \nu) = \sup_{\phi, \psi \in C^0} \langle \phi, \mu \rangle + \langle \psi, \nu \rangle - \varepsilon \langle e^{\frac{\phi \oplus \psi - c}{\varepsilon}}, \mu \otimes \nu \rangle$ is very popular in the OT community. We show in this section that the discrete potentials provided by the empirical Sinkhorn model can indeed be extended to continuous, convex and smooth potentials. However, their extension is not strongly convex and we prove that the semi-dual diverges almost surely when evaluted on empirical Sinkhorn potentials; hence we propose to quadratically regularize the Sinkhorn potentials. From a numerical point of view, we show how to compute efficiently the semi-dual on this regularized model with a fast converging scheme.

**A convex smooth model.** Recall that the first-order optimality condition on the Sinkhorn potential $\phi_\varepsilon$ gives $\phi_\varepsilon(x) = -\varepsilon \log(\int_y e^{\frac{\psi_\varepsilon(y) - c(x,y)}{\varepsilon}} d\nu(y))$. As done for instance in Berman [2018], Pooladian and Niles-Weed [2021], we use this conidtion to extend the empirical Sinkhorn potentials $(\hat{\phi}_\varepsilon, \hat{\psi}_\varepsilon) = \arg\min_{(\phi, \psi)} S_\varepsilon(\hat{\mu}, \hat{\nu})$ on the whole domain $\mathbb{R}^d$. Defining the associated Brenier potentials as $\hat{f}_\varepsilon = q - \hat{\phi}_\varepsilon$, we obtain $\hat{f}_\varepsilon(x) = \varepsilon \log(\int_y e^{\hat{\beta}_\varepsilon(y)} e^{\frac{x^\top y}{\varepsilon}} d\nu(y))$, where we defined $\hat{\beta}_\varepsilon(y) = e^{(1/\varepsilon)(\hat{\psi}_\varepsilon(y) - q(y))}$. This shows that the potentials $(\hat{f}_\varepsilon, \hat{g}_\varepsilon)$ are *Log-Sum-Exp* (LSE) functions and in particular, they are convex.

**Proposition 5.** *$\hat{f}_\varepsilon$ is $\frac{D^2(supp(\hat{\nu}))}{\varepsilon}$ smooth where $D(supp(\hat{\nu}))$ is the diameter of the support of $\hat{\nu}$.*

The proof is left in Appendix. While LSE functions are indeed smooth, they are not strongly convex. The following proposition shows that the semi-dual diverges almost surely on $\hat{f}_\varepsilon$.

**Proposition 6.** *Let $(\hat{f}_\varepsilon, \hat{g}_\varepsilon) = (q - \hat{\phi}_\varepsilon, q - \hat{\psi}_\varepsilon)$. If $\nu$ has continuous density with respect to the Lebesgue measure we have almost surely $\langle \hat{f}_\varepsilon^*, \nu \rangle = +\infty$.*

The proof is left in Appendix. This Proposition implies that evaluating the semi-dual on the empirical Sinkhorn potentials is ill-posed and it justifies their strong-convexity regularization. To this end, we consider $Q_\delta(\hat{f}_\varepsilon) = \hat{f}_\varepsilon + \delta q$ with $\delta > 0$ small and shall now read $J(Q_\delta(\hat{f}_\varepsilon)) - J_0 \leq \frac{1}{2\delta} e_\mu(\hat{f}_\varepsilon) + O(\delta)$. We show in the experiments of Sec. 5 that despite this small bias, the selection of empirical Sinkhorn potentials via the semi-dual criterion still provides satisfactory results.

**Self-concordant potentials.** Now, even if the Legendre transform can be computed via a standard convex first order minimization algorithm, it will not be effective in practice as the problem is conditioned by $O(\frac{1}{\delta \varepsilon})$. One way to circumvent poor conditioning is to employ second order methods. In the standard cases, they require costly line-searches however if the function has a *generalized self-concordant* structure, we can use a second order algorithm of the form $x_{k+1} = x_k - \alpha_k (\nabla^2 f(x_k))^{-1} \nabla f(x_k)$, where $\alpha_k$ is an explicit step size given in Sun and Tran-Dinh [2019, Theorem 2] and that provably yields a super-linearly convergent algorithm.

**Definition 1.** *Let $\alpha > 0$ and $f$ be a $C^3$ convex function. The function $f$ is said to be $(\alpha, M_f)$ self-concordant if for all $(x, u, v)$, $|(\nabla^3 f(x)[v]u)^\top u| \leq M_f \|u\|_x^2 \|v\|_x^{\alpha-2} \|v\|^{3-\alpha}$, where $\|u\|_x^2 =$*

$(\nabla^2 f(x)u)^\top u$, $\nabla^3 f(x)$ *is the tensor* $(\frac{\partial^3 f}{\partial x_i x_j x_k})_{1 \le ijk \le d}$ *and for a tensor* $T = (t_{ijk})_{1 \le i,j,k \le n}$, $T[v] = \sum_{i=1}^{p} v_i T_i$ *with* $T_i$ *the matrix* $(t_{ijk})_{1 \le j,k \le n}$.

Informally, the self-concordance measures how fast the Hessian varies with respect to the metric it induces.

**Proposition 7.** *The Sinkhorn Brenier potential* $\hat{f}_\varepsilon$ *is* $(2, \frac{D(\hat{\nu})}{\varepsilon})$ *self-concordant where the diameter* $D$ *is defined as* $D(\hat{\nu}) = \sup_{y \in \hat{\nu}, z \in \hat{\nu}} \|y - z\|_2$.

The proof is left in Appendix. Fig. 2 shows that the second order scheme is much faster to compute the Fenchel conjugate on $n = 1000$ points, with learning rate $O(\frac{1}{\varepsilon})$ for the first-order method.

# 5 Numerical experiments

We first introduce two other transport models on which we perform our numerical experiments.

## 5.1 Other models

**Input Convex Neural Network (ICNN).** The convex Brenier potentials are modeled by $(f_{\theta_1}, g_{\theta_2})$ two ICNN. Then the model is trained using a minimax objective $\min_{\theta_1} \max_{\theta_2} \langle f_{\theta_1}, \hat{\mu} \rangle + \frac{1}{n} \sum_{i=1}^{n} \nabla g_{\theta_2}(y_i)^\top y_i - \langle f_{\theta_1}, (\nabla g_{\theta_2})_{\#}(\hat{\nu}) \rangle$ The maximization aims at recovering $g_{\theta_2} \approx f_{\theta_1}^*$ and the minimization approximately solves the

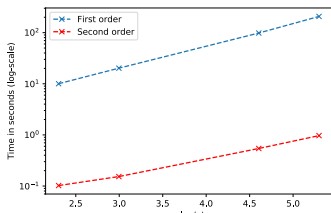

Figure 2: Fenchel Conjugate: 1st order vs 2nd order.

semi-dual. The implementation is based on the code of the authors[1]. Softplus activation layers were used instead of ReLu to obtain less degenerated maps. Note that since the weights $\theta_1, \theta_2$ are not controlled, we expect this model to provide lowly regular maps.

**Smooth Strongly convex Nearest Brenier (SSNB).** This model estimates the potential $f$ by approximately solving $\inf_{f \in \mathcal{F}_{(l,L)}} W_2((\nabla f)_{\#}(\hat{\mu}), \hat{\nu})$, where $\mathcal{F}_{(l,L)}$ is the set of $L$-smooth, $l$-strongly convex functions. As opposed to the previous model, SSNB provides maps that are very regular (in a bi-Lipschitz sense).

## 5.2 The experiments

**Synthetic XP.** We compare the ability of the models to recover the ground truth transport map using the semi-dual criterion map for three different distributions in a medium dimension setting $d = 8$. In all three cases, the distribution $\mu$ is uniform on the cube $[0,1]^d$ and $\nu$ is given by $(\nabla f)_{\#}(\mu)$ where $f$ is a convex function ; in virtue of Brenier's theorem, $T = \nabla f$ is the ground truth OT map between $\mu$ and $\nu$. The function $f$ has 3 different forms.

(i) Quadratic: $f(x) = \frac{1}{2} x^\top Q x + x^\top b$ where $Q = O^\top D O + 0.25 \, \mathrm{Id}$ where $O$ is a randomly chosen orthogonal matrix, $D$ is a random diagonal matrix whose entries are uniform in $[0,1]$ and $b$ is a random $d$-dimensional gaussian. This is a standard benchmark which simply aims at recovering a translation. (ii) Tensorized: $T_0(x) = x + (6 - \cos(6\pi x) - 0.2)^{-1}$ and $T(x) = \sum_{k=1}^{d} T_0(x_k)$. The map to learn is more complex but has a low dimensional structure as it pushes independently each directions. (iii) (Regularized) Log-Sum-Exp: $f(x) = t \, \mathrm{LSE}(\frac{C}{t} x + b) + \delta q(x)$ where the matrix $C$ is comprised of 10 centers uniformly chosen in $[-1,1]^d$, the shift $b$ is a random $d$-dimensional gaussian, the temperature $t$ was fixed at 0.3 and the regularizer $\delta = 0.001$. Note that any convex function can be approximated by such a Log-Sum-Exp [Calafiore et al., 2020]. However, because of this parametric structure, we expect the Sinkhorn model to be favored.

The training of the models, the semi-dual estimation and the Monte-Carlo approximation of the error error[2] $e_{\hat{\mu}}(f_i) = \int \|\nabla f_i(x) - T_0(x)\|^2 d\hat{\mu}(x)$ are done with 3 independent batches of size $n = 1024$. For the computation of the semi-dual, we regularized with $+\delta q$ the ICNN and Sinkhorn models taking $\delta = 1e - 3$. Note that the errors are computed on the original potentials, *without* the

---

[1] https://github.com/AmirTag/OT-ICNN
[2] Concentrates in $O(\frac{1}{\sqrt{n}})$ toward the "true" error.

| | ICNN | | Sinkhorn | | SSNB | |
|---|---|---|---|---|---|---|
| | $e_\mu(f_{\theta_{i_1}})$ | $e_\mu(f_{\theta_{i_0}})$ | $e_\mu(f_{\theta_{i_1}})$ | $e_\mu(f_{\theta_{i_0}})$ | $e_\mu(f_{\theta_{i_1}})$ | $e_\mu(f_{\theta_{i_0}})$ |
| Quad | 5.11 | 12.23 (16.13/48) | 0.036 | 0.047 (1.93/5) | 0.013 | **0.014** (1.33/11) |
| Tens | 2.74 | 2.74 (1.22/48) | 0.059 | 0.119 (2.72/5) | 0.006 | **0.006** (1.0/11) |
| LSE | 1.69 | 3.02 (17.11/48) | 0.006 | **0.006** (1.68/5) | 0.16 | 0.17 (1.26/11) |

Table 1: Parameter tuning for OT models. Each model is trained with different parameters $(\theta_i)_{1 \le i \le p}$ and a parameter $\theta_{i_0}$ is selected with the semi-dual criterion $\theta_{i_0} = \arg\min_i J(\hat{f}_{\theta_i})$. We report the error $e_\mu(f_{\theta_{i_0}})$ against $e_\mu(f_{\theta_{i_1}}) = \min_i e_\mu(f_{\theta_i})$; ideally $e_\mu(f_{\theta_{i_0}}) = e_\mu(f_{\theta_{i_1}})$. The numerator between brackets corresponds to the rank of the tuned potential with respect to the error $e_\mu$: the closer to one, the better. The denominator corresponds to the number of parameters. In bold the model with the best performance after being tuned with the semi-dual criterion.

regularization term. Forty-eight combinations of parameters were tested for the ICNN model, five for the Sinkhorn model and eleven for the SSNB model. More details on the parameters and on the implementation are given in Appendix.

The results were averaged on fifteen independent runs and are reported on Table 1. The parameters of each model are tuned with the semi-dual criterion . We denote $\theta_{i_0}$ the parameter that minimizes semi-dual and $\theta_{i_1}$ the parameter that minimizes the error $e_{\hat{\mu}}$. The numerator between brackets corresponds to the rank of the selected parameter with respect to the error $e_{\hat{\mu}}$; the closer to one the better. In particular, if $\theta_{i_0} = \theta_{i_1}$ we obtain the rank 1. The denominator is number of parameters: for instance, we tested the Sinkhorn model with five different $\varepsilon$, hence the denominator is 5.

| | ICNN/Sinkhorn/SSNB | |
|---|---|---|
| | $e_\mu(f_{i_1})$ | $e_\mu(f_{i_0})$ |
| Quad | 0.013 | 0.014 (1.33/64) |
| Tens | 0.006 | 0.006 (1.0/64) |
| LSE | 0.006 | 0.006 (1.68/64) |

Table 2: Model selection for OT: once the three models are tuned with the semi-dual, we select one among the three again with the semi-dual criterion. Note that the denominator is now equal to the sum of the number of parameters that we tested for each model.

In the case of SSNB where the smoothness and strong convexity parameters are explicitly controlled, the best parameter is almost always chosen. In the case of the Sinkhorn model, the regularity decreases for small values of $\varepsilon$ yet the selected potential remains in the top $40\%$ for the Quadratic and Log-Sum-Exp experiments. Conversely, the regularity is not controlled in the ICNN model yet the selected parameters remains in the top-tier for the Quadratic and Log-Sum-Exp experiments; as for the Tensorised experiment, the best potential is almost always selected.

Once the three models are calibrated with the semi-dual, we select one among them with the same criterion (see Table 2); note the denominator between brackets is now equal to the sum of the number of parameters considered for each model. We observe that the criterion selects the best or nearly-best transport map. This shows that the Brenier criterion can be both used for calibration and selection. Fig. 3 plots the semi-dual values against the error in the Log-Sum-Exp experiment and empirically suggests an even better behavior than best model selection. For the Sinkhorn and SSNB models, the error strictly increases with the semi-dual value, hence the semi-dual can not only select but can also directly rank the potentials with respect to the error $e_\mu$. For the ICNN model, we do not observe the same monotone behavior but we still get a positive correlation between the error and the semi-dual value. This less consistent behavior can be explained once again by the lack of control on the regularity of the potentials given by the model.

Overall, when we compare the models after being calibrated with the semi-dual we observe that ICNN always has the poorest performance. We may not have chosen the hyperpameters and the network structures in the best possible way and the ground truth may not favour this model. The SSNB model performs better by almost an order of magnitude than

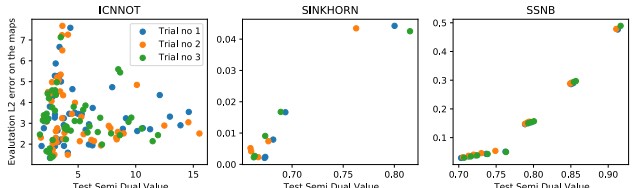

Figure 3: Empirical Semi-Dual against Quadratic Error on the LSE experiment. Ideally, the error should strictly increase with the semi-dual.

Sinkhorn on the Quadratic and Tensorized experiments. Conversely, as expected, the Sinkhorn model is the best one on the Log-Sum-Exp experiment. In terms of computation time, the training of the SSNB model takes between one and three hours and between 30 minutes and one hour for the semi-dual computation on a 120 GB RAM CPU. ICNN and Sinkhorn take a few minutes and a few seconds respectively for the training and semi-dual computation on a RTX6000 GPU.

Thanks to the high scalabilty of Sinkhorn, we repeated those three experiments on larger batches with $n = 10000$ averaged on 10 runs. As shown on Table 3, Sinkhorn recovers the same behavior as SSNB with respect to the semi-dual. Not only the best parameter $\varepsilon$ is always chosen but also, we show in Appendix that in the Quadratic and LSE settings, the error increases with the semi-dual.

|  | Quad | Tens | LSE |
|---|---|---|---|
| $e_\mu(f_{\varepsilon_1})$ | 0.0104 | 0.0376 | 0.0005 |
| $e_\mu(f_{\varepsilon_0})$ | 0.0104 | 0.0376 | 0.0005 |
| Rank | 1.0/5 | 1.0/5 | 1.0/5 |

Table 3: Parameter tuning for Sinkhorn model with $n = 10000$ points. The numerator of the Rank corresponds to the rank of the potential calibrated with the semi-dual criterion with respect to the error $e_\mu$: the closer to one, the better. The denominator corresponds to the number of parameters. We observe that the semi-dual accurately tunes the Sinkhorn model as the best parameter is always selected.

Before ending this paragraph, we briefly discuss the effect of the quadratic regularization for the ICNN and Sinkhorn models. We observed experimentally that without it, the Legendre transform indeed diverged over several points. On the other hand, the experiments did show that the semi-dual accurately selects the best potential even in the presence of the regularizer. We propose the following informal explanation: we show in Appendix that $J(Q_\delta(f)) - J_0 = \int Q_\delta(f)^*(T_0(x)) - Q_\delta(f)^*(T(x)+\delta x) + x^\top(T(x)+\delta x - T_0(x)) \, d\mu(x)$. We can decompose the integration on areas where the original potential $f^*$ is $O(1)$ Lipschitz, where we can expect $|f^*(T(x)) - Q_\delta(f)^*(T(x) + \delta x)| = O(\delta)$ and on the remaining area, the integrand behaves like $O(\frac{1}{\delta})$. For instance in the case of the empirical Sinkhorn model whose conjugate is Lipschitz on the interior of the convex hull of $\hat{\nu}$, we conjecture that $J(Q_\delta(\hat{f}_\varepsilon)) - J_0 \leq \mathrm{Cst}\sqrt{e_\mu(\hat{f}_\varepsilon)} + O(\delta + \frac{1}{\delta n^\alpha})$ where $\alpha > 0$ [3].

**Domain Adaptation. (DA)** The goal of DA is to infer unknown labels of a target distribution $X_t$ using a shifted source distribution $X_s$ with known labels $Y_s$. In the work of Courty et al. [2017] and many others [Redko et al., 2019, Xu et al., 2020], a map $T$ is sought between $X_s$ and $X_t$. Then a classifier $c$ is learned on $(T(X_s), Y_s)$ and is used to predict the unknown labels of the target as $c(X_t)$. Their core assumption is that $T$ should be close to the OT map between $X_s, X_t$. Question: is this assumption valid? Is the "true" OT map between $X_s$ and $X_t$ the one that will achieve the best knowledge transfer? Problem: among all proposed models, how to assess which map is the closest to the "true" *unknown* OT map? Using our criterion, we can now select the parameters and the model that will be closest to this ground truth.

We use the Caltech-office dataset which is a set of images of objects from ten distinct categories coming from four different sources of various quality: objects found in the online Amazon catalog (A), objects whose pictures have been taken with a webcam (W), with a high resolution digital SLR camera (D) and the Caltech-256 dataset (C) which is comprised of Google images. We use all the nine distinct pairs as source/domain data. As in Courty et al. [2017], in order for the quadratic distance to be meaningful, we do not use the raw images but feed them to a Decaf [Donahue et al., 2014] network and extract the features of the last layer and we use a 1-Nearest Neighbors as the classifier. The models and parameters we use are the same as in the Synthetic experiment. In our setting, the transport map $T$ is learned on train sets $X_s^{train}, X_t^{train}$ and the semi-dual is evaluated on test sets $X_s^{test}, X_t^{test}$, with 70% of the data for the train and 30% for the test.

The results are reported on Table 4 where only (A) and (W) are used as sources, the rest of the results are reported in Appendix and exhibit a similar pattern. We denote by $i_0$ the index of the potential minimizing the empirical semi-dual criterion and by $i_1$ the potential achieving the highest accuracy. The numerator between bracket corresponds to the rank of the selected potential with respect to the accuracy obtained when the classifier is learned on $(\nabla f(X_s), Y_s)$; the closer to 1, the better and

---

[3]It is shown in Brunel [2014, Equation (2.3)] that if $\mu$ has a convex support, the volume of $\mathrm{support}(\mu) \setminus \mathrm{Hull}(\hat{\mu})$ is upper-bounded by $O(n^{-\alpha})$. Depending on the shape of the support, it varies between $n^{-1}\log(n)^{d-1}$ and $n^{-2/(d+1)}$.

|       | ICNN |                | Sinkhorn |                | SSNB |                |
|-------|------|----------------|----------|----------------|------|----------------|
|       | $\mathrm{acc}(f_{i_1})$ | $\mathrm{acc}(f_{i_0})$ | $\mathrm{acc}(f_{i_1})$ | $\mathrm{acc}(f_{i_0})$ | $\mathrm{acc}(f_{i_1})$ | $\mathrm{acc}(f_{i_0})$ |
| A/C | 0.41 | 0.34 (2/48) | 0.84 | **0.82** (3/5) | 0.85 | 0.79 (10/11) |
| A/D | 0.44 | 0.15 (33/48) | 0.87 | 0.78 (4/5) | 0.82 | **0.8** (5/11) |
| A/W | 0.36 | 0.07 (48/48) | 0.78 | **0.72** (3/5) | 0.79 | 0.71 (9/11) |
| C/A | 0.47 | 0.09 (44/48) | 0.91 | 0.82 (5/5) | 0.91 | **0.88** (9/11) |
| C/D | 0.65 | 0.27 (6/48) | 0.9 | 0.8 (4/5) | 0.88 | **0.82** (10/11) |
| C/W | 0.36 | 0.34 (3/48) | 0.82 | 0.79 (2/5) | 0.83 | **0.83** (1/11) |

Table 4: Potential Selection for Domain-Adaptation. The column $\mathrm{acc}(f_{i_1})$ corresponds to the best (highest) accuracy and $\mathrm{acc}(f_{i_0})$ corresponds to the accuracy of the potential selected with the Brenier criterion. On this Table, the potentials are ranked with respect to the accuracy; the closer to one, the better the classification. In bold, the highest accuracy after being calibrated with the semi-dual.

in particular, $\mathrm{rank}(f_{i_1}) = 1$. We observe that the potential having the lowest accuracy is regularly selected by the semi-dual, even in the case of the SSNB model for which the semi-dual indicates very reliably the quality of the transport map. Hence we conclude that for DA, the best mapping for label transfer is not close to an optimal transport map. We remark that this conclusion is similar to the results of Korotin et al. [2021] and Stanczuk et al. [2021] who observed that the transport models which performed best for various ML tasks were not the ones that recover the sharpest OT maps.

## 6   Conclusion

The semi-dual Brenier formulation of quadratic OT provides us with a feasible criterion for convex potential selection. If the potentials are convex, this criterion can be computed numerically. Theoretically and experimentally, we showed that up to a sharp distortion parameter, the potential that minimizes the semi-dual is the one whose gradient minimizes the squared error to the ground truth map. Hence this criterion gives a fair and accurate procedure to benchmark convex OT models and provides a first step towards parameter tuning for stochastic OT. Regarding ML applications, this criterion allowed us to bring a first partial negative answer to the question of relevance of OT in DA: indeed, one may achieve good results in DA using models borrowed from OT literature, yet we claim that their performance is not correlated with how well they approximate the ground truth OT map between the source and the target. Possible extensions of our work could include more general cost $c$ and more general potentials. We believe $M$-smoothness and $\gamma$-strong convexity assumptions could be alleviated by a careful analysis of the quadratic and Moreau-Yosida regularizations. If not possible, the question of smoothness parameters selection in stochastic OT is widely open.

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
