# A ADDITIONAL PROOFS

## A.1 Proof of Lemma 1 (strongly convex case)

*Proof.* The major part of the proof is adapted from Muzellec et al. [2021, Lemma 3.1]. We denote $T = \nabla f$ and $T_0$ an OT map between $\mu$ and $\nu$ such that there exists a convex potential $f_0$ verifying $\nabla f_0 = T_0$. By definition of the Fenchel-Legendre transform, we have for any convex function $f$ and point $x$

$$f(x) + f^*(\nabla f(x)) = x^\top \nabla f(x), \tag{2}$$

Integrating this relation over $\mu$ for the optimal potential $f_0$ yields

$$\langle f_0, \mu \rangle + \langle f_0^* \circ T_0, \mu \rangle = \int x^\top T_0(x) \, d\mu(x). \tag{3}$$

Using the property $T_\#(\mu) = \nu$, we obtain that the r.h.s. is equal to $J_0$. We use this same property to re-write $J(f) = \int f(x) + f^*(T_0(x)) \, d\mu(x)$. Finally, using the Legendre identity stated above, we have $J(f) = \int f^*(T_0(x)) - f^*(\nabla f(x)) + x^\top \partial f(x) \, d\mu(x)$, which leads to

$$J(f) - J_0 = \int f^*(T_0(x)) - f^*(\nabla f(x)) - (T_0(x) - \nabla f(x))^\top x \, d\mu(x). \tag{4}$$

Recalling that $\partial f^*(\nabla f(x)) = x$, where $f^*$ is a subgradient of $f^*$, we identify in the integrand a Bregman divergence $D_{f^*}(T_0(x), \nabla f(x))$ where for a convex function $h$ the Bregman divergence $D_h(y, x) = h(y) - h(x) - \partial h(x)^\top(y - x)$. When $f$ is assumed $\gamma$-strongly convex, $f^*$ is $\frac{1}{\gamma}$-smooth and $D_{f^*}(T_0(x), \nabla f(x))$ is upper-bounded by $\frac{1}{2\gamma}\|T(x) - T_0(x)\|^2$ which yields

$$J(f) - J_0 \leq \frac{1}{2\gamma} \int \|T(x) - T_0(x)\|^2 \, d\mu(x). \tag{5}$$

Conversely, when $f$ is assumed $M$-smooth, $f^*$ is $\frac{1}{2M}$-strongly convex and $D_{f^*}(T_0(x), \nabla f(x))$ is lower-bounded by $\frac{1}{2M}\|T(x) - T_0(x)\|^2$ which yields

$$J(f) - J_0 \geq \frac{1}{2M} \int \|T(x) - T_0(x)\|^2 \, d\mu(x). \tag{6}$$

$\square$

## A.2 Proof of Prop. 1

*Proof.* Define the potential $g_0(x) = |x| + \frac{x^2}{2}$ and for $0 \leq \lambda \leq \frac{1}{2}$, define the translated potential $g_\lambda = g_0(\cdot - \lambda)$. Let us start by computing the Legendre transform of $g_0$. The Legendre transform of $g_0$ is defined for all $y$ as

$$g_0^*(y) = \sup_{x \in \mathbb{R}} xy - g_0(x). \tag{7}$$

Since $g_0(x) \geq |x|$ for all $x$, then if $y \in [-1, 1]$ then $g_0^*(y) = 0$. If $y > 1$, since $g_0$ is pair and positive, the maximum is attained on $\mathbb{R}^+$. Denoting $\phi(x, y) = xy - g_0(x)$, we have that $\phi(\cdot, y)$ increases between $[0, y-1]$ and decreases between $[y-1, +\infty[$ hence the maximum is attained in $x = y - 1$ which yields $g_0^*(y) = \frac{(y-1)^2}{2}$. Conversely, if $y < -1$ we have $g_0^*(y) = \frac{(y+1)^2}{2}$. From this result, we can compute $g_\lambda^*$ in virtue of the relation $g_\lambda^*(y) = g^*(y) + \lambda y$.

Let us now compute the semi-dual $J(g_\lambda)$. The first term is given by

$$\int_{-\frac{1}{2}}^{\frac{1}{2}} g_\lambda(x)\,\mathrm{d}x = \int_{-\frac{1}{2}}^{\lambda} g(x-\lambda)\,\mathrm{d}x + \int_{\lambda}^{\frac{1}{2}} g(x-\lambda)\,\mathrm{d}x \tag{8}$$

$$= \int_{-\frac{1}{2}}^{\lambda} \frac{(x-\lambda)^2}{2} + (\lambda - x)\,\mathrm{d}x + \int_{\lambda}^{\frac{1}{2}} \frac{(x-\lambda)^2}{2} + (x-\lambda)\,\mathrm{d}x \tag{9}$$

$$= [\frac{(\cdot-\lambda)^3}{6}]_{-\frac{1}{2}}^{\frac{1}{2}} + [-\frac{(\cdot-\lambda)^2}{2}]_{-\frac{1}{2}}^{\lambda} + [-\frac{(\lambda-\cdot)^2}{2}]_{\lambda}^{\frac{1}{2}} \tag{10}$$

$$= \frac{(\frac{1}{2}-\lambda)^3 + (\frac{1}{2}+\lambda)^3}{6} + \frac{(\frac{1}{2}+\lambda)^2 + (\frac{1}{2}-\lambda)^2}{2} \tag{11}$$

$$= \frac{1}{4} + \frac{\frac{1}{4}+3\lambda^2}{6} + \lambda^2\,. \tag{12}$$

The second term is given by

$$\int_{\mathbb{R}} g_\lambda^*(y)\,\mathrm{d}(\nabla g_0)(\mu)(y) = \int_{-\frac{1}{2}}^{\frac{1}{2}} g_\lambda^*(\nabla g_0(y))\,\mathrm{d}y \tag{13}$$

$$= \int_{-\frac{1}{2}}^{0} g_0^*(y-1-\lambda)\,\mathrm{d}y + \int_0^{\frac{1}{2}} g_0^*(y+1-\lambda)\,\mathrm{d}y \tag{14}$$

$$= \int_{-\frac{1}{2}} \frac{(u-\lambda)^2}{2}\,\mathrm{d}u + \int_{\lambda}^{\frac{1}{2}} \frac{(u-\lambda)^2}{2}\,\mathrm{d}u \tag{15}$$

$$= [\frac{(\cdot-\lambda)^3}{6}]_{-\frac{1}{2}}^{0} + [\frac{(u-\lambda)^3}{6}]_{\lambda}^{\frac{1}{2}} \tag{16}$$

$$= \frac{(\frac{1}{2}+\lambda)^3 + (\frac{1}{2}-\lambda)^3 - \lambda^3}{6} \tag{17}$$

$$= \frac{\frac{1}{4}+3\lambda^2 - \lambda^3}{6}\,. \tag{18}$$

Hence the semi-dual is given by $J(g_\lambda) = \frac{1}{3} + 2\lambda^2 - \lambda^3$. Finally, let us compute the error $e_\mu$

$$e_\mu(g_\lambda) = \int_{-\frac{1}{2}}^{0} (\nabla g_0(x) - \nabla g_\lambda(x))^2\,\mathrm{d}x + \int_0^{\lambda} (\nabla g_0(x) - \nabla g_\lambda(x))^2\,\mathrm{d}x + \int_{\lambda}^{\frac{1}{2}} (\nabla g_0(x) - \nabla g_\lambda(x))^2\,\mathrm{d}x \tag{19}$$

$$= \frac{\lambda^2}{2} + \lambda(2+\lambda)^2 + \lambda^2(\frac{1}{2}-\lambda) \tag{20}$$

$$= 4\lambda + 5\lambda^2\,. \tag{21}$$

$\square$

## A.3 Proof of Prop. 2

*Proof.* We begin with splitting $\hat{J}(f_{i_0}) - J_0$ in non-stochastic and stochastic terms

$$\hat{J}(f_{i_0}) - J_0 = J(f_{i_0}) - J_0 + \hat{J}(f_{i_0}) - J(f_{i_0})\,, \tag{22}$$

where we denoted $\hat{J}$ the empirical semi-dual $J_{\hat{\mu},\hat{\nu}}$ Using Lemma 1, we get the lower bound

$$\hat{J}(f_{i_0}) - J_0 \geq \frac{1}{2M} e_\mu(f_{i_0}) + \hat{J}(f_{i_0}) - J(f_{i_0})\,. \tag{23}$$

By construction, $f_{i_0}$ verifies for all $1 \leq i \leq p$

$$\hat{J}(f_{i_0}) - J_0 \leq \hat{J}(f_i) - J_0$$
$$= J(f_i) - J_0 + \hat{J}(f_i) - J(f_i)\,.$$

Picking $i = i_1$ and using Lemma 1, we obtain

$$\hat{J}(f_{i_0}) - J_0 \leq \frac{1}{2\gamma} e_\mu(f_{i_1}) + \hat{J}(f_{i_1}) - J(f_{i_1}). \tag{24}$$

Equations (23) and (24) give

$$e_\mu(f_{i_0}) \leq \frac{M}{\gamma} e_\mu(f_{i_1}) + 2M(\hat{J}(f_{i_1}) - J(f_{i_1})) \tag{25}$$

$$+ 2M(J(f_{i_0}) - \hat{J}(f_{i_0})). \tag{26}$$

The Hoeffding lemma gives for all $t > 0$

$$\mathbb{P}(\langle f_i, \hat{\mu} - \mu \rangle \geq t) \leq \exp\left(-\frac{2nt^2}{\|f_i\|_{osc,X}^2}\right). \tag{27}$$

We place ourselves on the event

$$A = (\langle f_{i_1}, \hat{\mu} - \mu \rangle \geq t) \cup (\langle f_{i_1}^*, \hat{\nu} - \nu \rangle \geq t) \\ \cup (\langle f_{i_0}, \mu - \hat{\mu} \rangle \geq t) \cup (\langle f_{i_0}^*, \nu - \hat{\nu} \rangle \geq t). \tag{28}$$

We want to set $\mathbb{P}(A) \leq \delta$. By triangle inequality, we get the upper-bound

$$\mathbb{P}(A) \leq 4\exp\left(-\frac{2nt^2}{C^2}\right), \tag{29}$$

where $C = \max(C_{i_0}, C_{i_1})$ and $C_i$ defined as $C_i = \max(\|f_i\|_{X,o}, \|f_i^*\|_{Y,o})$. Hence setting, $t = C\sqrt{\frac{\ln(4/\delta)}{2n}}$, we have with probability at least $1 - \delta$

$$e_\mu(f_{i_0}) \leq \frac{M}{\gamma} e_\mu(f_{i_1}) + 8MC\sqrt{\frac{\ln(4/\delta)}{2n}}. \tag{30}$$

$\square$

## A.4 Proof of Prop. 3

*Proof.* Take $\mu \sim [0,1]$ and $f_0 \equiv 0$ and let us compute the error for $g = M\frac{x^2}{2}$.

$$e_\mu(g) = \int_0^1 (Mx)^2 \, \mathrm{d}x \tag{31}$$

$$= \frac{M^2}{3}. \tag{32}$$

Conversely, defining $h_\epsilon = \gamma\frac{x^2}{2} + (\epsilon + \alpha_{M,\gamma})x$ with

$$\alpha_{M,\gamma} = \frac{\gamma}{2}\left[\sqrt{1 + \frac{4(M-\gamma)}{3\gamma}} - 1\right], \tag{33}$$

the error $e_\mu(h_\epsilon)$ is given by

$$e_\mu(h) = \int_0^1 (\gamma x + (\epsilon + \alpha_{M,\gamma}))^2 \, \mathrm{d}x \tag{34}$$

$$= \frac{\gamma^2}{3} + \gamma(\alpha_{M,\gamma} + \epsilon) + (\alpha_{M,\gamma} + \epsilon)^2 \tag{35}$$

$$= \frac{\gamma^2}{3} + \gamma\epsilon + \gamma\alpha_{M,\gamma} + \epsilon^2 + 2\epsilon\alpha_{M,\gamma} + \alpha_{M,\gamma}^2 \tag{36}$$

$$= \frac{\gamma^2}{3} + \gamma\epsilon + \frac{\gamma^2}{2}\sqrt{1 + \frac{4(M-\gamma)}{3\gamma}} - \frac{\gamma^2}{2} + \epsilon^2 + 2\epsilon\alpha_{M,\gamma} + \frac{\gamma^2}{4}\left[2 + \frac{4(M-\gamma)}{3\gamma} - 2\sqrt{1 + \frac{4(M-\gamma)}{3\gamma}}\right] \tag{37}$$

$$= \frac{\gamma^2}{3} + \gamma\epsilon + \epsilon^2 + 2\epsilon\alpha_{M,\gamma} + \frac{\gamma(M-\gamma)}{3} \tag{38}$$

$$= \frac{M\gamma}{3} + \gamma\epsilon + \epsilon^2 - \gamma\epsilon + \gamma\epsilon\sqrt{1 + \frac{4(M-\gamma)}{3\gamma}} \tag{39}$$

$$= \frac{M\gamma}{3}\left[1 + \frac{3\epsilon^2}{M\gamma} + \frac{3\epsilon}{M}\sqrt{1 + \frac{4(M-\gamma)}{3\gamma}}\right]. \tag{40}$$

In particular, we obtain $\frac{e_\mu(g)}{e_\mu(h_\epsilon)} = \frac{M}{\gamma} \times \frac{1}{1 + \frac{3\epsilon}{M}\left[\frac{\epsilon}{\gamma} + \sqrt{1 + \frac{4(M-\gamma)}{3\gamma}}\right]} \xrightarrow{\epsilon \to 0} \frac{M}{\gamma}$. Now, let us compute the

semi-duals $J(g)$ and $J(h_\varepsilon)$. Since $f_0 \equiv 0$, we have $\nu = \delta_0$ a Dirac mass in $0$. Hence we simply
need to compute the Legendre transform of $g$ and $h_\epsilon$ in $0$

$$g^*(0) = \sup_x \; -M\frac{x^2}{2} \tag{41}$$

$$= 0 \,, \tag{42}$$

and

$$h_\varepsilon^*(0) = \sup_x \; -\gamma\frac{x^2}{2} - (\epsilon + \alpha_{M,\gamma})x \tag{43}$$

$$= \frac{(\epsilon + \alpha_{M,\gamma})^2}{2\gamma} \,. \tag{44}$$

Hence we obtain

$$J(g) = \int_0^1 M\frac{x^2}{2} \, \mathrm{d}x \tag{45}$$

$$= \frac{M}{6} \,, \tag{46}$$

and

$$J(h_\epsilon) = \int_0^1 \gamma\frac{x^2}{2} + (\epsilon + \alpha_{M,\gamma})x \, \mathrm{d}x + \frac{(\epsilon + \alpha_{M,\gamma})^2}{2\gamma} \tag{47}$$

$$= \frac{\gamma}{6} + \frac{\epsilon + \alpha_{M,\gamma}}{2} + \frac{(\epsilon + \alpha_{M,\gamma})^2}{2\gamma} \tag{48}$$

$$= \frac{\gamma}{6} + \frac{\epsilon + \alpha_{M,\gamma}}{2} + \frac{\epsilon^2 + 2\epsilon\alpha_{M,\gamma} + \alpha_{M,\gamma}^2}{2\gamma} \tag{49}$$

$$= \frac{1}{2\gamma}\left(\frac{\gamma^2}{3} + \gamma(\epsilon + \alpha_{M,\gamma}) + \epsilon^2 + 2\epsilon\alpha_{M,\gamma} + \alpha_{M,\gamma}^2\right). \tag{50}$$

We recognize between brackets the same expression as $e_\mu(h_\epsilon)$ in Equation (52) hence we obtain

$$J(h_\epsilon) = \frac{M}{6}\left[1 + \frac{3\epsilon^2}{M\gamma} + \frac{3\epsilon}{M}\sqrt{1 + \frac{4(M-\gamma)}{3\gamma}}\right]. \tag{51}$$

$\square$

## A.5 Proof of Prop. 4

*Proof.* Applying Lemma 1, we have for all $\delta > 0$ the inequality $J(Q_\delta(f)) - J_0 \leq \frac{1}{2\delta} e_\mu(Q_\delta(f))$. The right hand side is decomposed in $\langle Q_\delta(f), \mu \rangle + \langle Q_\delta(f)^*, \nu \rangle$. The first term is simply $\langle f, \mu \rangle + \delta \langle q, \mu \rangle$. For the second-term we use the standard result $Q_\delta(f)^* = M_\delta(f)$ the Moreau-Yosida transform of $f$ reading $M_\tau(f) = \inf_y f(y) + \frac{q(x-y)}{\tau}$. If $f^*$ is L-Lipschitz on the support of $\nu$, we have the lower bound $M_\delta(f) \geq f - \frac{L^2\delta}{2}$. Hence we recover

$$J(f) - J_0 - \frac{L^2\delta}{2} + \delta\langle q, \mu \rangle \leq \frac{1}{2\delta} e_\mu(Q_\delta(f)) \,. \tag{52}$$

The term $e_\mu(Q_\delta(f))$ is upper-bounded by $2(e_\mu(f) + 2\delta^2\langle q, \mu \rangle)$ which gives $J(f) - J_0 \leq \frac{e_\mu(f)}{\delta} + \delta\langle q, \mu \rangle + \frac{L^2\delta}{2}$. Optimizing on $\delta$ leads to

$$J(f) - J_0 \leq 2\sqrt{e_\mu(f)\left(\frac{L^2}{2} + \langle q, \mu \rangle\right)} \,. \tag{53}$$

$\square$

## A.6 Proof of Prop. 5

*Proof.* Recall that the Fenchel-Legendre of a standard Log-Sum-Exp function $\text{LSE}(x) = \log(\sum_{i=1}^n e^{x_i})$ is given by

$$\text{LSE}^*(y) = \sum_{i=1}^n y_i \log(y_i) + \iota(y \in \mathcal{S}_n) \tag{54}$$

$$= -\text{Ent}(y) + \iota(y \in \mathcal{S}_n) \,, \tag{55}$$

where $\mathcal{S}_n$ is the probability simplex. More generally, defining $\text{LSE}_b(x) = \log(\sum_{i=1}^n e^{x_i + b_i})$, using the fact that $f^*(\cdot + \tau) = f^*(\cdot) - \tau^\top \cdot$, we have

$$(\text{LSE}_b)^*(y) = -\text{Ent}(y) - b^\top y + \iota(y \in \mathcal{S}_n) \,. \tag{56}$$

At the optimum, for empirical measures $\hat{\mu} = \frac{1}{m} \sum_{i=1}^m \delta_{x_i}, \hat{\nu} = \frac{1}{n} \sum_{i=1}^n \delta_{y_i}$ the empirical Sinkhorn Kantorovitch potentials $(\hat{\phi}_\varepsilon, \hat{\psi}_\varepsilon)$ are linked as

$$\hat{\phi}_\varepsilon(x) = -\varepsilon \log\left(\frac{1}{n} \sum_{i=1}^n e^{2\frac{2\hat{\psi}_\varepsilon(y_i) - \|x - y_i\|^2}{2\varepsilon}}\right) \,, \tag{57}$$

hence the Sinkhorn Brenier potential $f_\varepsilon$ can be written as

$$f_\varepsilon(x) = \varepsilon \, \text{LSE}_{b_\varepsilon}(C_\varepsilon x) \,, \tag{58}$$

where we defined

$$\begin{cases} C_\varepsilon = (\frac{y_i}{\varepsilon})_{1 \leq i \leq n} \in \mathbb{R}^{n \times d} \\ b_{\varepsilon,n} = (\frac{2\psi_\varepsilon(y_i) - \|y_i\|^2}{2\varepsilon} - \log(n))_{1 \leq i \leq n} \in \mathbb{R}^n \end{cases} \,. \tag{59}$$

Now recall that

- $(\varepsilon f(\cdot))^* = \varepsilon f^*(\frac{\cdot}{\varepsilon})$.

- $\forall z, (f(A\cdot))^*(z) = \inf_{Ay=z} f^*(y)$.

Hence we can deduce

$$f_\varepsilon^*(y) = \varepsilon \inf_{C\Delta = y} -\text{Ent}(\Delta) - \Delta^\top b_{\varepsilon,n} + \iota(\Delta \in \mathcal{S}_n) \,,$$

where $C \in \mathbb{R}^{n \times d}$ is the matrix of the samples $(y_i)$. In particular if $f_\varepsilon^*$ is evaluated outside the convex hull of $\hat{\nu}$, it is infinite. Since $\nu$ has continuous density, there almost surely exists $(y_0, r), r > 0$ such that $B(y_0, r) \subset Supp(\nu)$ and $B(y_0, r) \cap Conv(\hat{\nu}) = \emptyset$ where $Conv(\hat{\nu})$ is the convex hull of the samples $\hat{\nu}$. In particular, almost surely

$$\langle f_\varepsilon^*, \nu \rangle = +\infty \,. \tag{60}$$

$\square$

## A.7 Proof of Prop. 7

The proof is largely inspired from an article on the online blog of Francis Bach[4].

Since the 2-self-concordance is scaling invariant, we shall simply prove that $f(x) = \mathrm{LSE}_b(C.)$ is $(2, D(C))$ self-concordant with $b \in \mathbb{R}_+^n$, $C \in \mathbb{R}^{n \times d}$ the matrix whose rows are centers $(c_i)_{1 \leq i \leq n}$ and $D(C) = \max_{ij} \|c_i - c_j\|$.

*Proof.* Defining the (non-normalized) distribution $\zeta = \frac{1}{n} \sum_{i=1}^n b_i \delta_{c_i}$, we can remark that $f$ is the normalizing factor of the conditional exponential distribution

$$h(c|x) \propto e^{c^\top x} d\zeta(c) \tag{61}$$

$$= e^{c^\top x - f(x)} d\zeta(c). \tag{62}$$

The gradient of $f$ is given by

$$\nabla f(x) = \frac{\int c e^{c^\top x} d\zeta(c)}{\int e^{c^\top x} d\zeta(c)} \tag{63}$$

$$= \mathbb{E}_h(c), \tag{64}$$

and using the results of Pistone and Wynn [1999], we have for higher order derivatives

$$\nabla^p f(x) = \mathbb{E}_h(\otimes_{j=1}^p (c - \nabla f(x))), \tag{65}$$

where for a vector $v \in \mathbb{R}^d$, $\otimes_{j=1}^p v$ is a tensor $V_p$ in $\mathbb{R}^{d^p}$ whose entries are $(v_{i_1} \times \cdots \times v_{i_p})$. In particular, applying the formula for $p = 3$ and denoting $H = (c - \nabla f(x)) \otimes (c - \nabla f(x))$

$$\nabla^3 f(x) = \mathbb{E}_h[(c - \nabla f(x)) \otimes H]. \tag{66}$$

Using the linearity of the expectation, we have

$$|(\nabla^3 f(x)[v]u)^\top u| = |\mathbb{E}_h[(c - \nabla f(x))^\top v \times (Hu)^\top u]| \tag{67}$$

$$\leq \mathbb{E}_h[|(c - \nabla f(x))^\top v| \times |(Hu)^\top u|]. \tag{68}$$

Since $\nabla f(x) \in Conv(C)$, we have in particular that $\|c - \nabla f(x)\| \leq D(C)$. Furthermore since $H$ is a positive matrix, we obtain the following upper-bound

$$|(\nabla^3 f(x)[v]u)^\top u| \leq D(C)\|v\|\mathbb{E}_h[(Hu)^\top u] \tag{69}$$

$$\leq D(C)\|v\|(\nabla^2 f(x)u)^\top u. \tag{70}$$

$\square$

## A.8 Proof of Prop. 5

*Proof.* The Sinkhorn Brenier empirical potentials are of the form $f_\varepsilon = \varepsilon \, \mathrm{LSE}_{b_{\varepsilon,n}}(C_\varepsilon.)$ where $C_\varepsilon$ and $b_{\varepsilon,n}$ are defined in (59). Using the formulas from the previous proof, we simply have to bound $H_{c,x} = (c - \nabla f(x)) \otimes (c - \nabla f(x))$

$$u^\top H_{c,x} u = (u^\top (c - \nabla f(x)))^2 \tag{71}$$

$$\leq \|u\|_2^2 \|c - \nabla f(x)\|_2^2. \tag{72}$$

Since $\nabla f(x)$ is in the convex hull of $\frac{\hat{\nu}}{\varepsilon}$ and $c \in Supp(\frac{\hat{\nu}}{\varepsilon})$, we deduce that $\|H_{c,x}\|_{op} \leq \frac{D^2(\hat{\nu})}{\varepsilon^2}$, where $\|.\|_{op}$ is the spectral norm. In particular $\|\nabla^2 f(x)\|_{op} \leq \frac{D^2(\hat{\nu})}{\varepsilon}$. $\square$

# B MISCELLANEOUS

## B.1 DA experiment

We present here the results in the Domain Adaptation experiment where the source terms are (D) and (W) respectively. The results are displayed on Table 5: again, the best accuracy for the downstream classification task is not correlated with the minimization of the semi-dual, in particular the best OT maps are not suited for label transfer.

---

[4]https://francisbach.com/self-concordant-analysis-for-logistic-regression/

|  | ICNN | | Sinkhorn | | SSNB | |
|---|---|---|---|---|---|---|
|  | $\mathrm{acc}(f_{i_1})$ | $\mathrm{acc}(f_{i_0})$ | $\mathrm{acc}(f_{i_1})$ | $\mathrm{acc}(f_{i_0})$ | $\mathrm{acc}(f_{i_1})$ | $\mathrm{acc}(f_{i_0})$ |
| D/A | 0.5 | 0.47 (2/48) | 0.91 | 0.78 (5/5) | 0.91 | **0.84** (11/11) |
| D/C | 0.54 | 0.43 (2/48) | 0.83 | 0.74 (5/5) | 0.83 | **0.75** (9/11) |
| D/W | 0.52 | 0.28 (11/48) | 0.96 | 0.85 (4/5) | 0.99 | **0.95** (8/11) |
| W/A | 0.48 | 0.25 (17/48) | 0.89 | **0.78** (4/5) | 0.87 | 0.77 (11/11) |
| W/C | 0.4 | 0.2 (13/48) | 0.77 | 0.73 (4/5) | 0.78 | **0.74** (10/11) |
| W/D | 0.62 | 0.51 (2/48) | 0.95 | 0.9 (3/5) | 1.0 | **1.0** (1/11) |

Table 5: Potential Selection for Domain-Adaptation. The column $\mathrm{acc}(f_{i_1})$ corresponds to the best (highest) accuracy and $\mathrm{acc}(f_{i_0})$ corresponds to the accuracy of the potential selected with the Brenier criterion. On this Table, the potentials are ranked with respect to the accuracy; the closer to one, the better the classification. In bold, the highest accuracy after being calibrated with the semi-dual.

## B.2 SSNB algorithm

For $l < L$, the SSNB model is defined as

$$\inf_{f \in \mathcal{F}_{l,L}} W_2^2((\nabla f)_\#(\mu), \nu), \tag{73}$$

where $\mathcal{F}_{l,L}$ is the set of $l$-strongly convex, $L$-smooth functions. For empirical potentials $\hat{\mu} = \frac{1}{n}\sum_{i=1}^n \delta_{x_i}$ and $\hat{\nu} = \frac{1}{m}\sum_{i=1}^m \delta_{y_i}$, the authors propose to solve the non-convex problem (73) in an alternate fashion: for a fixed $f \in \mathcal{F}_{l,L}$, they estimate the transport coupling $(P_{ij}) \in \mathbb{R}^{n \times m}$ from $(\nabla f)_\#(\hat{\mu})$ to $\hat{\nu}$ by solving the associated linear program (or an entropic approximation) and then, once the coupling is fixed, they estimate $f$ (pointwise on $\hat{\mu}$) by solving

$$\min_{(z_1, \cdots, z_n) \in \mathbb{R}^{n \times d}, u \in \mathbb{R}^n} \sum_{ij} P_{ij} \|z_i - y_j\|_2^2$$

$$\text{subject to } u_i \geq u_j + z_j^\top (x_i - x_j) + \frac{1}{2(1 - l/L)} \left( \frac{1}{L} \|z_i - z_j\|^2 + \frac{1}{l} \|x_i - x_j\|_2^2 - \frac{2l}{L}(z_j - z_i)^\top (x_i - x_j) \right), \tag{74}$$

where $z_i = \nabla f(x_i)$ and $u_i = f(x_i)$. The problem above is a convex Quadratically Constrained Quadratic Problem and can be numerically solved with CVXPY for instance. However, when such an option is chosen the $n(n-1)$ constraints must be computed at each iterations which induces a large overhead. Instead, we reformulate this problem as a standard linear conic problem of the form $Ax - b \in \mathcal{K}$, with $\mathcal{K}$ a fixed cone to be compiled only once.

**From QCQP to SOCP** First we show how to reformulate a (convex) QCQP without equality constraints into an SOCP. The standard formulation of a QCQP is

$$\inf_x \ \frac{1}{2} x^\top Q_0 x + c_0^\top x$$
$$\text{s. t. } \frac{1}{2} x^\top Q_i x + c_i^\top x + r_i \leq 0, \ i = 1, \cdots, p. \tag{75}$$

Introducing the slack variables $(t_0, t_1, \cdots, t_p) = \frac{1}{2}(x^\top Q_0 x, x^\top Q_1 x, \cdots, x^\top Q_p x)$, we re-write the problem as

$$\inf_{x,t} \ t_0 + c_0^\top x$$
$$\text{s. t. } t_i + c_i^\top x + r_i = 0, \ i = 1, \cdots, p$$
$$t_i \geq \frac{1}{2} x^\top Q_i x, \ i = 0, \cdots, p. \tag{76}$$

Decomposing $Q_i$ as $Q_i = F_i^\top F_i$ with $F_i$ having $p$ rows, the constraint $t_i = \frac{1}{2} x^\top Q_i x$ becomes $(1, t_i, F_i x) \in \mathcal{Q}_r^{d+2}$, where $\mathcal{Q}_r^{d+2}$ is the rotated $(d+2)$-dimensional Lorentz cone defined as

$$\mathcal{Q}_r^{d+2} = \{(x_1, x_2, \cdots, x_{d+2}) \text{ s.t. } 2x_1 x_2 \geq \sum_{k=1}^{d} x_{i+2}^2\}. \tag{77}$$

We obtain a MOSEK-friendly formulation of the QCQP as

$$\begin{aligned}
\inf_{x,t} \quad & t_0 + c_0^\top x \\
\text{s. t. } & t_i + c_i^\top x + r_i = 0, \quad i = 1, \cdots, p \\
& (1, t_i, F_i x) \in \mathcal{Q}_r^{d+2}, \quad i = 0, \cdots, p,
\end{aligned} \tag{78}$$

which has the form $Ax - b \in \mathcal{K}$ where $\mathcal{K}$ is a fixed product of Lorentz cone whose number and dimensions solely depend on $n$ and $d$ in the case of SSNB. Hence we can compile $\mathcal{K}$ only once for fixed $(n, d)$, which allows us to considerably reduce the overhead.

**Decomposition of $Q_{ij}$**   In the SSNB model the symmetric positive matrices $Q_{ij} \in \mathcal{S}_{n(d+1)}^+(\mathbb{R})$ are defined up to a common scaling parameter as

$$\begin{cases}
q_{kl} = 1 \text{ if } k = l \in \{di, \cdots (d+1)i\} \cup \{dj, \cdots (d+1)j\} \\
q_{kl} = -1 \text{ if } l = k + dj, k \in \{di, \cdots (d+1)i\} \\
q_{kl} = -1 \text{ if } k = l + dj, l \in \{di, \cdots (d+1)i\}.
\end{cases} \tag{79}$$

The matrix $Q_{ij}$ is factorized as $F_{ij}^\top F_{ij}$ with $F_{ij} \in \mathbb{R}^{d \times n(d+1)}$ defined as

$$\begin{cases}
f_{kl} = 1 \text{ if } l = k + di, k \in \{1, \cdots, d\} \\
f_{kl} = -1 \text{ if } l = k + dj, k \in \{1, \cdots, d\}.
\end{cases} \tag{80}$$

### B.3   Models parameters

**ICNN**   We used a 3-layers ICNN with softplus activations. The number of hidden neurons was chosen in $\{64, 128, 256\}$, the soft convexity penalty for the potential $g$ and the matching moment/variance penalty were both chosen in $\{0, 0.001, 0.01, 0.1\}$. As recommended by the authors, the batch size was set to 60, the number of epochs was set to 60, the number of inner iterations to approximate the conjugate was set to 25 and the learning rate is initially set to 1e-4 and is then divided by 2 every 2-epochs.

To compute the semi-dual, we regularized the potential $f$ by adding $\frac{\delta}{2}\|x\|^2$ with $\delta =$ 1e-3. The numerical optimization was done with SciPy with a stopping condition set to $0.001$ ; for a lower stopping criterion, the minimization would not converge.

**Sinkhorn**   The temperature $\varepsilon$ was chosen in $\{0.5, 0.1, 0.05, 0.01, 0.005\}$. We stopped the training when the optimality conditions are almost met

$$\begin{cases}
\langle |\phi_\varepsilon(.) + \varepsilon \log(\int_y e^{\frac{\psi_\varepsilon(y) - c(., y)}{\varepsilon}} d\hat{\nu}(y))|, \hat{\mu} \rangle \leq \text{1e-5} \\
\langle |\psi_\varepsilon(.) + \varepsilon \log(\int_x e^{\frac{\phi_\varepsilon(y) - c(x, .)}{\varepsilon}} d\hat{\mu}(x))|, \hat{\nu} \rangle \leq \text{1e-5}.
\end{cases} \tag{81}$$

The resulting Sinkhorn Brenier potential $\hat{f}_\varepsilon$ is regularized with $\frac{\delta}{2}\|x\|^2$, $\delta = 0.001$. When the semi-dual is computed on a point $y_i$, the stopping criterion is given by

$$\|\nabla \hat{f}_\varepsilon(z_t) - y_i\| \leq \text{1e-5}, \tag{82}$$

where $z_t$ is the current point of the optimization at time step $t$.

**SSNB**   The strong convexity parameter $l$ is chosen in $\{0.2, 0.5, 0.7, 0.9\}$ and the smoothness parameter $L$ is chosen in $\{0.2, 0.5, 0.7, 0.9, 1.2\}$ with $l < L$. The number of iterations in the alternate minimization is set to 10. The conjugate is computed with a first order scheme with learning rate $\frac{1}{2L}$ and is stopped with the same criterion as above.

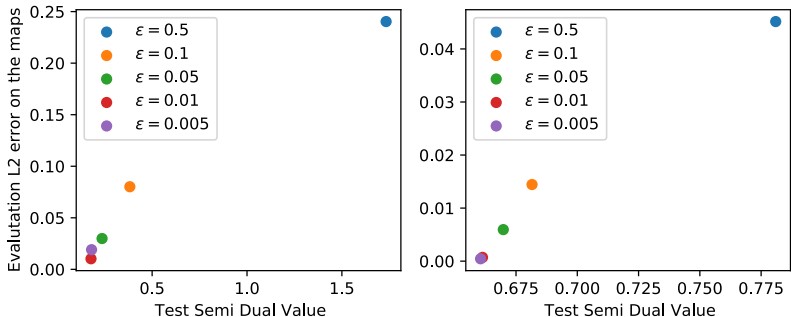

Figure 4: Empirical Semi-Dual against Quadratic Error on the Quadratic and Log-Sum-Exp experiments for the Sinkhorn model, $n = 10000$ and $d = 8$.

### B.4 Additional Experiment Sinkhorn

We run 10 times the Quadratic and Log-Sum-Exp experiments with the Sinkhorn model but on $n = 10000$ points for the training of the model, the semi-dual and the computation of the error. The results are reported on Figure B.4. Just as for SSNB, the semi-dual can accurately rank the potentials according to their error $e_\mu(f_i) = \int \|\nabla f_i(x) - T_0(x)\|_2^2 d\mu(x)$ where $T_0$ is the ground truth OT map.