# OpenReview forum: "Parameter tuning and model selection in Optimal Transport with semi-dual Brenier formulation"
_NeurIPS.cc/2022/Conference — NeurIPS 2022 Accept_

### Official Review · Reviewer_vPxL · 2022-06-27

**Rating:** 5
**Confidence:** 4
**Soundness:** 2 fair
**Presentation:** 2 fair
**Contribution:** 2 fair

**Summary:**

The paper proposes a sample-based criterion to select the best convex Brenier potential among the list of available ones for the Wasserstein-2 optimal transport. The criterion is based on the theoretical result (lemma 1) which connects the semi-dual Brenier formulation of OT and the L2 error between the gradient of the approximate potential and the OT map. The main idea of the criterion is to simply take the potential for which the empirical semi-dual is the smallest: for this potential, one may guarantee that its gradient will not be far from the best available OT map (Proposition 2). The authors provide some quantitative evaluation in small dimensional spaces to illustrate that the potential selected by their criterion is good (they use SSNB, ICNN-based and Sinkhorn algorithms to get the candidate potentials). Also, they conduct a domain adaptation experiment to show that that the improved accuracy of computing the OT may does not necessarily yield increase in DA accuracy metric.


**Questions:**

Incorporated to the previous part.

**Note:** My assessment is mostly based on the main text. I have looked through the main text + briefly looked through appendices. I did not run the code of the supplementary and did not check the proofs of appendices.

**Limitations:**

There is no separate section of paragraph with limitations, but some of them are discussed in the text, e.g., the necessity to consider smooth/strongly convex potentials.

Another important limitation, which seems not to be discussed in the paper, it the fact that the proposed criterion significantly relies on the brenier OT formulation, convexity of the dual potentials, Fenchel transform, etc. Thus,
(1) the same analysis/ideas would presumably not generalize to other OT transport cost functions beyond W2 (quadratic cost);
(2) the application of the criterion requires convex potentials which are in practice are either ICNNs or LSE functions (in the Sinkhorn case), which are both very restrictive. It is unclear whether these architectures will perform perform well in practical tasks.

It would be great if the authors could carefully discuss these important limitations in the answers and speculate on possible ways to overcome them.

**Post-rebuttal:** I increase my score from 4 to 5.

**Strengths And Weaknesses:**

**Strength.** To my knowledge, there is no closely research related to model selection for optimal transport. The current paper seems to be the first who studies this question and proposes an approach to model selection in W2 optimal transport. It derives an upper bound on the empirical semi-dual error and provides an example showing that it is tight (in the non-stochastic regime).

The domain adaptation experiment also raises an interesting and important question: is it reasonable to apply OT in DA problems? Emphasizing and discussing this question might be interesting to the ML/OT community. The current paper shows that better DA performance not necessarily correlates with the OT performance, which points to a gap between computational OT and downstream applications.

**Weaknesses.** I think the paper has several important flaws.

*(1) Clarity.* The overall exposition, writing and presentation could be improved. First, most of the equations are in-line formulas which makes it hard to read the paper. Next, there is simply no introduction to the optimal transport problem: an unprepared reader without significant experience in OT (especially in W2 OT related to the Brenier formulation) will struggle to understand the message of this paper. In the paper, the OT problem has never been formally defined, only its dual is considered. Due to this, it might be hard to understand how all these derivations are related to the domain adaptation problem studied later (why the gradient of the OT potential is a meaningful mapper for DA problem?). In general, it is not trivial to follow the stream of thoughts of the authors: there is limited discussion of the meaning of propositions/theoretical results and transition between them. For example, in my view, the main criterion is not clearly stated: how to apply it in practice? (Note: this could be understood from the experimental section, but I think the authors should devote a separate paragraph for this to make it clear).

*(2) Main contribution.* While the idea is novel, but one might argue that it is rather straightforward from the theoretical point of view. Indeed, the key proposition 2 follows from lemma 1 (which is a already known result in W2 optimal transport) and Hoeffding lemma (the authors themself note this in line 168).

*(3) Related work.* Most of the related work seems to be resonably covered. However, in some cases whether the provided results are new or they already exist. For example, as a reader, I do not understand whether results in section 4 are novel or it is just a discussion of some related OT background with illustrations. In the discussion of the related methods 246-252, it is not straightforward to come up with the optimization objective of line 248. The authors could have repeated a reference to the original method’s paper. Same applies to SSNB method (253-256). How to optimize this objective? Additional details and clarifications could be useful (in the main text (!), not appendix). Besides, there is a work [1] which is not cited in the current paper although it seems to be closely related as its authors also discuss the necessity to use strongly convex/smooth potentials (which is heavily relied on the current paper). Also, lemma 1 (which the authors use) is also present in [1].

[1] Korotin, A., Egiazarian, V., Asadulaev, A., Safin, A., & Burnaev, E. (2021). Wasserstein-2 Generative Networks. In International Conference on Learning Representations.

*(4) Experiments.* The synthetic experiments seem to be a little bit limited. I think the authors should consider more examples and in different dimensions (not only d=8) to study the practical usefulness of their criterion. In the current form, the evaluation (Tables 2, 1, 3) is not sufficiently convincing.

Relevance of DA experiments. As I noted earlier, the OT DA part of the paper raises and discusses an interesting question: is the OT map a valid adaptation map in DA? While the question is important, I would argue that the answer to the question is just obvious: for some dataset pairs OT map may be a meaningful, for some - not. In particular, I have checked three papers that the authors cite [Courty et al., 2017], [Redko et al., 2019], [Xu et al., 2020]. If I understand correctly, the methods are mostly discrete (they are not interested in the dual potentials as functions contrary to the current work) and their authors typically propose costs/regularization to make the mapping obtain desired properties instead of  using the “raw” W2 cost (which is studied here) [between features]. Thus, the DA experiment in the current paper is a nice illustration of their criterion but is note very significant.

To conclude, I think the most serious weaknesses are (1) [clarity] and (4) [experimental evaluation].

---

> ### Author Response · Authors · 2022-08-01
> **Response vPxL**
>
> Dear reviewer, we believe your critics are mainly founded on an over interpretation of our claims which are 1) our criterion provides an accurate way to rank potentials with respect to $e_\mu$ 2) “raw” quadratic OT should not be used for domain adaptation
>
> ### Weakness
> (1) Clarity
> We acknowledge that for non OT practitioners, the paper may appear technical. However, we assume the topic of the paper should mainly interest the OT community. Yet we shall take into account your remarks in the final version.
>
> (2) Main Contribution
> Indeed the main contribution of the paper is not a technical one (even though the Propositions 1, 3 and 4 are new results and non completely trivial). Still, the question of OT model selection is relevant ; we simply exhibited a criterion that provably and empirically does a good job at it. We believe that makes it a legit contribution by being a new application of a previously known tool to an important question.
>
> (4) Experiments
> *“authors should consider more examples [...] not sufficiently convincing.”*
> We acknowledge we could take as a ground truth ICNN models, in order to further explore the limiting case of non-smooth and non-strongly convex settings. We could also change the dimension but it simply would make the learning task harder, not necessarily the ranking task which is our only focus.
> However, we believe that Tables (2, 1, 3) are sufficiently convincing at showing that the semi-dual properly selects the best OT model.
>
> *“If I understand correctly, the methods [in the OT based DA papers] are mostly discrete (they are not interested in the dual potentials as functions contrary to the current work)”*
> This statement is not correct: even though they use discrete methods, they implicitly use the underlying dual potentials. In [Courty et. al. 2017] they compute the coupling matrix between the source and the target $\Gamma \in \mathbb{R}^{n \times m}$. Then each element of the source $x_i^{s}$ is sent on the target with a barycentric mapping $z_{i} = \sum_{j} \Gamma_{ij} x_{j}^{t}$ where $x_{j}^{t}$ is an element of the target. In the case for instance of a Sinkhorn coupling, one has exactly
> $$
> z_i = \nabla \hat{f}_\varepsilon(x_i^{s})
> $$
> where $ \hat{f}_\varepsilon$ is the associated Sinkhorn potential. Hence they implicitly use the dual potentials when they map the source on the target.
>
> *“their authors typically propose costs/regularization to make the mapping obtain desired properties instead of using the “raw” W2 cost”*
> The purpose of the study is precisely to show that “raw” OT mapping does not work.
>
> ### Limitations
> (1) *“Work does not generalize to other costs”*
> Formally, it does generalize to $W_1$, in this case the semi-dual is $J(f) = \langle f, \mu - \nu \rangle$ (which one should maximize instead of minimize) and can be computed for non-convex functions.
> Conversely, the work also formally generalizes to costs that are convex in their first arguments and concave in their second arguments. In this case, if we restrict ourselves to $c$-concave potentials (note that the true OT potential is always $c$-concave, see for instance [1]), then the $c$-transform $\min_{y} c(x, y) - f(y)$ is a convex problem and can be computed.
> We do not know if stability results could also be derived though.
>
> (2) *“Only works for convex functions [...]. It is unclear whether these architectures will perform well on practical tasks”*
> The question of which architecture performs best on a specific ML task is not our concern. Our only concern is to select which structure best approximates the ground truth OT map. Note that since at the optimum, the OT potential is convex, the structure should indeed plug convexity in. Yet we acknowledge that one may want to approximate a convex function with a non-convex model and this is a limitation of our work.
> There are two main issues when using non convex functions, which can be overcome in some cases:
>
> (i) The computation of the Legendre transform $\max_x x^\top y - f(x)$ must be done numerically up to a high precision. If not, then we underestimate $J(f)$ making $f$ look a better candidate than what it is. If we are in a small dimension (1, 2 or 3), we may use a sufficiently precise grid and get a fair estimate of $J(f)$ without assuming convexity.
>
> (ii) We may not get stability results as the ones provided in Lemma 1 without assuming convexity. However, the lower bound $\frac{1}{2M}e_\mu(f) \leq J(f) - J(f_0)$ still holds when $f$ is $M$-smooth but not necessarily convex. Furthermore, as shown in Prop 4, if $f^*$ is $L$ lipschitz on the support of $\nu$, we can recover the upper bound $J(f) - J(f_0) \leq 2\sqrt{e_\mu(f)(L^2 + \langle q, \mu \rangle)}$ where $q(x) = \frac{\|x\|^2}{2}$ ; again we do not need convexity here. This inequality is weaker but it is sufficient to obtain guarantees for model selection.
>
> [1] Filippo Santambrogio, Optimal Transport for Applied Mathematicians. In Birkäuser NY, 2015.

---

> > ### Comment · Reviewer_vPxL · 2022-08-05
> > **Response to rebuttal**
> >
> > I thank the authors for clarifying some of my concerns about their paper. I understand that that this mostly theoretical/conceptual work and it may give some insights for further studies of optimal transport model selection. I agree with the authors' position that provided experiments are sufficient to support the main claims.
> >
> > However, since the work studies the question of model selection, which seems to be very practical, I wonder to which problems the developed criterion might be applied. Could the authors please further elaborate, are there any problems to which OT is applied and "better OT map/potential => better solution of the problem", i.e., selecting the best potential is indeed justified? The current work shows this implication is not true for domain adaptation and selecting the best potential is not very reasonable. If so, where it is reasonable and what are the potential applications of the criterion that the authors propose?

---

> > > ### Author Response · Authors · 2022-08-07
> > > **Response vPxL**
> > >
> > > Dear reviewer,
> > >
> > > Thank you for taking into account our response. Concerning your question *what can this criterion be useful for?*, we shall answer in three points:
> > >
> > > (1) we believe it should useful for the statistical OT community, for benchmarking purposes: assume you are a statistician and you design some estimator $f_\lambda$ of the OT potentials where $\lambda$ is some parameter and you want to know if your model is better than Sinkhorn $f_\varepsilon$. You generate some training data to train your model and Sinkhorn for different parameters, you compute the semi-dual on test data for each $\varepsilon$ and $\lambda$, you take for Sinkhorn $\varepsilon$ that minimizes this semi dual and for your model the $\lambda$ that minimizes this semi dual. Finally you compute the error for the two models on independent data and you observe which model is the best.
> > > This whole procedure provides a way to fairly compare models between each other. Before that, one could only compute the error for different $\lambda$ and $\varepsilon$ and one would usually some that for some $\lambda$ and $\varepsilon$ we had $e_\mu(f_\lambda) <e_\mu(f_\varepsilon)$ and for some, we had  $e_\mu(f_\lambda) > e_\mu(f_\varepsilon)$ which makes it hard to decide objectively which one is the best. This use case was the main motivation to write our paper.
> > >
> > > (2) just as we did for domain adaptation, the criterion can be used to determine whether OT is truly relevant for some applications. Given the current "hype" of OT in the ML community that can sometimes be unjustified, we believe it would be healthier to have this kind of tool.
> > >
> > > (3) we believe the criterion could be useful for time interpolation purposes ; for the sake of concreteness, we give here one example in computational biology: in [1], the RNA cell expression profile is taken at different time steps $(t_i)$ giving (empirical) genes expression distributions $\mu_{t_i}$ (that are embedded in $\mathbb{R}^d$). Then, these distributions are interpolated for any $t \in [t_i,t_{i+1}] $ using an OT model between $(\mu_{t_i}, \mu_{t_{i+1})$ ; we believe our criterion could be used to pick the best OT model, leading to smoother interpolations. Unlike the DA case, where the OT map introduction was ad-hoc, the OT map is explicitly searched here. We plan to benchmark this potential application in future work.
> > >
> > > [1] Geoffrey Schiebinger, Jian Shu, Marcin Tabaka, Brian Cleary, Vidya Subramanian, Aryeh Solomon, Joshua Gould, Siyan Liu, Stacie Lin, Peter Berube, et al. Optimal-transport analysis of single-cell gene expression identifies developmental trajectories in reprogramming. In Cell 2019.

---

> > > > ### Comment · Reviewer_vPxL · 2022-08-07
> > > > **Response**
> > > >
> > > > The authors have answered my questions and I increase the score to borderline accept.

---

### Official Review · Reviewer_Visd · 2022-07-01

**Rating:** 7
**Confidence:** 4
**Soundness:** 4 excellent
**Presentation:** 3 good
**Contribution:** 3 good

**Summary:**

This paper proposes a novel criterion to evaluate how close an estimated (L2 Wasserstein) optimal transport plan between two given measures is close to the true (unknown) optimal transport plan. To do so, the authors rely on the semi-dual formulation of the Kantorovich problem, and prove that the objective function for this formulation (using discrete measures with n points) is highly correlated with the L2 distance between the estimated transport plan and the true one (and give bounds). The authors require that the potential to be estimated has a Lipschitz gradient, and the true potential is strongly convex. When those criteria are met, the semi-dual objective is finite and can be used as a proxy for the distance to the optimal map. The authors show that the map minimizing this criterion are related to the distance between the true map and the closest one, through an inequality involving a factor depending on the regularity of the estimated map. When the regularity criteria are not met, the authors argue that the potentials can be regularized using an additive quadratic term for the semi-dual objective to be computable. The criterion can be used for hyperparameter tuning in various OT algorithms (including Sinkhorn iterations, that do not natively satisfy the requirements), or to evaluate whether performance for a downstream task (here domain adaptation) correlates or not with OT performance. Somewhat surprisingly, performance on a DA task does not correlate with the quality of the OT map.

**Questions:**

- It would be quite informative to check the bound validity on practical cases where everything is known (e.g OT between gaussians, where gamma and M, and T0 can be explicitly computed, or even the synthetic experiments with a quadratic potential). Examples in 1D such as the one illustrated in Fig. 1 (which is never really discussed beyond the introduction, though it brings the point across nicely) could also have been explored in more detail.

- On a related note, the bound provided in equation one is particularly interesting, since even if the function minimizing the semi-dual criterion has a small smoothness coefficient (M), the ratio M/gamma cannot become smaller than one (in the non-stochastic case). This would contradict the optimality of the model in the RHS. Are there results on this or do the authors have some insight on the interplay between the smoothness of f0 and the strong convexity of f1 ? Would it be for example possible to have an idea of the optimal model if only those constants are known, without computing the semi-dual criterion ?

- Why does the number of candidate models vary so much from one model to the next ? There are only 5 hyperparameter values for the Sinkhorn potentials, while the method is quite fast to train and the number of candidate models could have been higher.

- I checked [Caffarelli 2000] but could not find exactly what result was being referenced in theorem 1, but I could have missed something. Can the authors provide the exact reference to the theorem and its proof? Besides, the conclusion that under the hypotheses of this theorem, the Brenier potentials are smooth and strongly convex are not obvious to me and could deserve some explanation (the theorem only states they are convex and C2). Also, what happens for continuous measures with non compact support (e.g. Gaussians) ?

- The counter examples (propositions 1 and 3) are a bit abrupt and would benefit a bit more contextualization. For example, proposition 3 aims at showing in a practical case that the bound is tight in the infinite samples regime. But isn’t this a general result, valid for any case as long as n goes to infinity ? I am not sure what the example brings exactly.

**Limitations:**

The authors point out that the conditions required for the semi dual to be both finite and reliable in terms of correlating with the OT performance may be restrictive in some cases.
For Sinkhorn models, they resolve this in practice via quadratic regularizations and suggest useful bounds could be obtained under less strong regularity conditions.
They also show experimentally that the critetion may not correlated with OT performance when the regularity of the obtained potentials cannot be controlled (as in ICNNs).

**Strengths And Weaknesses:**

Strengths :

- The authors propose a conceptually simple way to rank candidate estimated OT maps in terms of how well they approximate the true map, without using the latter. This could be potentially very useful to compare algorithms in the future, or to check on pratical cases whether or not optimal transport is a good constraint to add to improve performance on a given downstream task.

- The theoretical developments are well presented and the empirical results are well illustrated by the experimental results. Namely the regularity in practice of the potentials is well shown (on simple computable examples and when comparing the different algorithms) to be crucial for the proposed semi-dual criterion to be computable and more importantly reliable.

- The approach can be applied in practical case where the semi-dual would normally not be useful (infinite) thanks to small regularizations providing the necessary smoothness and strong convexity. This in particular makes the approach applicable (and tractable thanks to efficent ways to compute the Legendre transform) to Sinkhorn models which are among the most common.

- The Domain Adaptation experiment provides interesting nsight on this problem and answers at least partially to the question of how useful OT is in this situation.

Weaknesses :

- The presentation and motivation foir presenting some of the propositions and theorems could be improved (See below for details on this point)

- The results on domain adaptation, given their potential consequences on the applicability of OT to this problem, would probably have to be confirmed on another case study, either on another real dataset, or maybe a low dimensional toy dataset (where maybe the optimal transformationis knwon) for this result to be strengthened.

---

> ### Author Response · Authors · 2022-08-01
> **Response Visd**
>
> Dear reviewer. We thank you for your nice and constructive feedback.
>
> ### Q1
> Do you suggest that we compute $(J(f_a) _ J(f_0))/e_\mu(f)$ when $\mu$ and $\nu$ are two normalized gaussians with different means and $f_a$ is for instance a potential of the form $\| x \|^2/2 +a x$? Indeed, it would be nice to see how far or close we are from the worst case bound. We can add this in Supplementary.
>
> ### Q2
> We are not sure that we have well understood the question. If you reformulate, we will try to answer.
>
> ### Q3
> Indeed the number of candidates vary a lot from one model to another because of the number of hyperparameters each model has. For each of these hyperparameters, we used a fairly coarse grid (about 5 different values of each parameter of each model). In particular, Sinkhorn has only one hyperparameter, SSNB has two (the smoothness and strong convexity) and ICNN has three (see Appendix for more details) hence the large discrepancy. Yet we believe that in the case of Sinkhorn, benchmarking 5 different $\varepsilon$ (with logarithmic spacing) already provides a fair panel of the model.
>
> ### Q4
> The precise reference is [1] Theorem 2b.
> This theorem gives that both the transport maps are C1 and in particular that the associated potentials are C2. Now, since we are on a compact set, they can be both made uniformly smooth. Since they are mutual Legendre transform of one another and that the Legendre transform of a smooth function is strongly convex, we have that both potentials are smooth and strongly convex. We shall make this reasoning explicit in the final version.
> The generic unbounded case is more technical yet we believe there exists references covering this case.
>
> ### Q5
> We believe the bound
> $$
> \frac{1}{2M} e_\mu(f)  \leq J(f) - J(f_0) \leq \frac{1}{2\gamma} e_\mu(f)
> $$
> is indeed tight for $f$ $M$-smooth, $\gamma$-strongly convex and that is not a new result (yet we did not find clear examples in the literature). Yet Prop. 1 and 3 do not aim to prove this tightness.
>
> The goal of Prop. 1 was to prove that the smoothness assumption is necessary to lower bound the quantity $J(f) - J(f_0)/e_\mu(f) $ and conversely the strong convexity is necessary to upper-bound the quantity  $J(f) - J(f_0)/e_\mu(f) $.
> The goal of Prop. 3 was to show that we could indeed exhibit $f_0$ and $f_1$ such that $J(f_0) < J(f_1)$ (or casually speaking, $f_0$ is a better candidate than $f_1$) and yet the error achieved by $f_0$ is worse by a $\frac{M}{\gamma}$ factor than the error achieved by $f_1$. This result is new.
> We shall make these explanations more explicit in the final version.
>
> [1] Luis A. Caffarelli. Monotonicity Properties of Optimal Transportation and the FKG and related Inequalities. In Communication in Mathematical Physics 2000.

---

> > ### Comment · Reviewer_Visd · 2022-08-05
> > **Reply to authors**
> >
> > Thanks for your reply, some points I raised are more clear to me now.
> >
> > Q1: Yes this is what I had in mind. It could be made for Gaussians with different means and even different covariance matrices.
> >
> > Q2: My point was that for functions that are both strongly convex and smooth, the associated constants provide a bound on the eigenvalues of its Hessian from both sides. This means that the ratio $M/\gamma$ cannot become smaller than 1. So I was wondering if one could exploit this knowledge, but it was irrelevant because $\gamma$ is the strong convexity constant of $f_0$, and $M$ the strong smoothness constant of the candidate model $f$ (so there is no a priori constraints linking both constants). The only thing we can say is that the true potential $f$ cannot have $M>\gamma$. My mistake.
> >
> > Q4: Thanks, I think it would be nice indeed to state this in the paper.

---

### Official Review · Reviewer_Qbsh · 2022-07-10

**Rating:** 6
**Confidence:** 4
**Soundness:** 2 fair
**Presentation:** 3 good
**Contribution:** 2 fair

**Summary:**

Given two measures $\mu$ and $\nu$, and a set of estimated potentials (from the dual formula in Optimal Transport theory) between them, the paper provides a criterion to select the potential having a corresponding pushforward map (derivative) being closest to the Monge map. Thanks to the assumptions of strong convexity and smoothness of the potentials, and Bernstein-typed of inequalities, we can provide the theoretical bounds for error of this method non-asymptotically (Lemma 1 and Proposition 2). Some experiments and applications are provided to support the method and theory. It also gives a good negative finding to Optimal Transport Domain Adaptation models that the best Domain Adaptation map is not close to the Monge map in several applications.

**Questions:**

1. The calculation in Proposition 1 seems to be not correct. Because in the OT dual formula, the constraint $\phi(x) + \psi(x) \leq (x-y)^2/2$ only needs to satisfy in the $Support(\mu)\times Support(\nu)$. Similar to constraint of the potentials $f(x) + g(y) \geq xy$. Therefore, in this case, because both the support of $\mu$ and $\nu$ are compact, we can not use the definition of Fenchel-Legendre on the whole domain $\mathbb{R}$ (but with respect to $[-1/2, 1/2]$ only). This potentially relates to other results and proofs in this paper.

2. Similar to the synthetic experiment, the Domain Adaptation experiment also needs to run with a few replications. The result should be then more reliable. When we learn the transport map $T$ from training data and evaluate potentials on the test data, how can we make sure that the distribution of the training and test data are the same? The experiment description should be more clear in this aspect.

**Ethics Review Area:**

["I don’t know"]

**Limitations:**

While the research question is interesting, the approach is good and natural, it appears to have several points that can be improved in this paper, both in terms of theory and experiments (see above). I suggest the authors examine those points carefully and make the paper better.

**Strengths And Weaknesses:**

**Strength**:

1. The idea of this paper is natural and good. From a naive perspective, it is natural to select the potential as the one optimizing the dual formulation. The theory from this paper supports this viewpoint, and the experiments empirically show it is effective. The take-away lesson in the paper is valuable to Optimal Transport and Domain Adaptation researchers.

2. The presentation of the paper is clear, the story is well-motivated, and the experiments part is carefully carried out.

**Weakness**:

1. There are still many typos in the manuscript. For example, in the caption of Figure 1: "pic", in line 208: "conidtion", and in line 209: "argmin" (instead of argmax).

2. Some of the arguments and theoretical results appear to be not correct (see Questions).

3. To make the presentation of the paper more fluent, I think the authors should merge Section 4 and Section 5.1 because they both talk about finding potentials. The paragraph "Self-concordant potentials" is not clearly written about how we can obtain the conjugate via the second-order method. It is better to just present the method to find the conjugate and a theoretical result about its sample complexity (combination of Definition 1 and Proposition 7) instead of introducing a new notion (generalized self-concordant). A detailed explanation can be put in the appendix.

---

> ### Author Response · Authors · 2022-08-01
> **Response Qbsh**
>
> Dear Reviewer. We thank you for your constructive feedback. Thank you for pointing out our typos, we shall correct them for the final version.
>
> Weakness 3.
> The reason we dedicated a separated section to the Sinkhorn model is because it is a very popular model and, fortunately, we can in this case efficiently compute the semi dual even if this model is non-strongly convex. We decided not to merge this Section with 5.1 because for the other models, it is straightforward to see than they indeed provide convex potentials that are well amenable to the computation of the semi-dual (for SSNB at least which provide well-conditioned potentials) while it is non trivial for Sinkhorn.
> We acknowledge that the description of how we obtain the conjugate in the self concordant paragraph may lack clarity as it is only briefly mentioned in l. 228. We shall make it more explicit in the final version.
> We did not understand the remark on how Definition 1 + Proposition 7 relates to a notion of sample complexity. Do you suggest that we should state something like: “ For Sinkhorn, the Fenchel Transform can be computed in $O(...)$ steps?”
>
> ### Q1
> *“Proposition 1 looks false”*
> In your opinion what precise statement or equality is false ?
>
> *“ the Legendre transform should not be taken over $\mathbb{R}$”*
> The Legendre transform can indeed be taken over $\mathbb{R}$: since $\mathbb{R}$ is a Polish space and the quadratic cost is bounded from below, the constraint $\phi \oplus \psi \leq \frac{\| x - y \|^2}{2}$ holds on the whole domain $\mathbb{R}$ (see [1] Theorem 1. 42).
>
> ### Q2
> *“More experiments should be made both on the synthetic and real-world data”*
> For the synthetic data, we acknowledge we could have added an experiment where the ground truth is an ICNN or an SSNB potential and maybe we could have tried with another dimension. We may add this in the supplementary for the final version. Conversely, we may add another case study with real-world data for the DA XP.
>
> *“How can we make sure the distribution of the test data and the train data is the same?”*
> Given some source distribution $X_s$ and target distribution $X_{tar}$, we randomly take 70% of $X_s$ as our training source data and the remaining as our test source data .The same is done for the target. Please note that the overall training data $X^{train}$ is comprised of the pair $(X_s^{train}, X_{tar}^{train})$ and that the overall test data $X^{test}$ is comprised of the pair $(X_s^{test}, X_{tar}^{test})$. We shall explicitly mention this construction in the final version.
> We believe it is reasonable to say that built as such, the training data and the test data follow the same distribution.
>
> [1] Filippo Santambrogio, Optimal Transport for Applied Mathematicians. In Birkäuser NY, 2015.

---

> > ### Comment · Reviewer_Qbsh · 2022-08-05
> > **Proposition 1**
> >
> > Hi, in Proposition 1, you did not provide proof for your first claim: "Take $\mu \sim U[-1/2, 1/2]$ and $f_0$ of the form $\lambda q(x) + x$ with $\lambda \geq 0$. The potential $f(x) = x$ is indeed convex with Lipschitz gradient and is such that $J(f) = \infty$ yet $e_{\mu}(f) = \frac{\lambda^2}{4}\to 0$". This claim is very questionable: both $\mu$ and $\nu$ has finite compact support with nice pdf's, but $J$ equals to $\infty$? I think it is likely because of the Legendre transform. Please let me know if I do not understand it correctly, or there is somewhere you provided the proof that I am not aware of.

---

> > > ### Author Response · Authors · 2022-08-07
> > > **Reponse Qbsh**
> > >
> > > Dear reviewer.
> > >
> > > Indeed, the semi-dual $J(f)$ diverges because the Legendre transform is taken over the whole domain as it should be. We do not really understand what is bothering you. Indeed, it may sound bothering from a practical point of view: the candidate $f(x) = x$ seems like a nice candidate yet it is heavily sanctioned by the semi-dual ; it was the goal of this proposition to show that if the strong convexity assumption is removed, the semi-dual may not be the right criterion to use. However, quadratic regularization can help as highlighted in Proposition 4 and in the discussion in Sec 5 around Sinkhorn model.

---

> > > > ### Comment · Reviewer_Qbsh · 2022-08-07
> > > > **Proposition 1**
> > > >
> > > > Hi, thank you for your answer. I checked the proposition carefully and I understand my mistake here. I increased the score to 6.

---

### Official Review · Reviewer_8oqQ · 2022-07-11

**Rating:** 6
**Confidence:** 5
**Soundness:** 3 good
**Presentation:** 2 fair
**Contribution:** 3 good

**Summary:**

This paper uses Lebesgue transform to give the semi-dual Brenier objective as the quantitative criterion for convex potential selection .
This criterion enables parameter tuning and model selection among entropic regularization of OT, input convex neural networks(ICNN) and smooth and strongly convex nearest-Brenier (SSNB) models.
The authors also use this criterion to question the use of OT in Domain-Adaptation.

**Questions:**

How to ensure that the final model selection is accurate if the performance of ICNN is not optimal?

**Limitations:**

They proposed the limitation of their algorithm, but doesn't address the potential negative societal impact .

**Strengths And Weaknesses:**

Strengths: this paper uses Lebesgue transform to give the semi-dual objective as the quantitative criterion . The theoretical derivation and experiments show the effectiveness of the method.
This paper has been proved by experiment that the map minimizing the semi-dual, hence the closest to the ground truth OT map, is often far from being the one that achieves the best label transfer.
Weakness: (Page 8,paragraph 2)The authors claimed the choice of network structure may lead to poor ICNN performance,which needs more experimental verification.

---

> ### Author Response · Authors · 2022-08-01
> **Response 8oqQ**
>
> Dear Reviewer. We thank you for your constructive feedback.
> *“How to ensure the final model selection is optimal if the performance of the ICNN is not optimal ?”*
> Our model selection is useful to discriminate between candidate models that are proposed. Our criteria will be applied to the model at hand, whether it is optimal or not in its class, such as ICNN. For the experiments, we cannot guarantee optimality of the selected ICNN model, however, we applied standard procedures to find it, as were proposed in the literature.

---

### Meta-Review · Area_Chair_JMFX · 2022-08-26

**Recommendation:** Accept
**Confidence:** Certain

**Metareview:**

While there were few misunderstandings, the rebuttal successfully convinced all the reviewers that the paper should be accepted. Please take into account the reviewers' comments in preparing the camera-ready, especially the ones concerning the clarity of the paper.

**Award:**

No

---

### Decision · Program_Chairs · 2022-09-14

Accept